# Mutationally-activated PI3'-kinase-α promotes de-differentiation of lung tumors initiated by the BRAF[V600E] oncoprotein kinase

J Edward van Veen[1,2,3,4†], Michael Scherzer[1,2], Julia Boshuizen[3,4‡], Mollee Chu[3,4§], Annie Liu[1,2], Allison Landman[1,4#], Shon Green[3,4¶], Christy Trejo[3,4**], Martin McMahon[1,2,3,4*]

[1]Huntsman Cancer Institute, University of Utah, Salt Lake City, United States; [2]Department of Dermatology, University of Utah, Salt Lake City, United States; [3]Helen Diller Family Comprehensive Cancer Center, University of California, San Francisco, San Francisco, United States; [4]Department of Cellular & Molecular Pharmacology, University of California, San Francisco, San Francisco, United States

*For correspondence:
martin.mcmahon@hci.utah.edu

Present address: †Department of Integrative Biology and Physiology, Jonsson Comprehensive Cancer Institute, University of California, Los Angeles, Los Angeles, United States; ‡Division of Molecular Genetics, Netherlands Cancer Institute, Amsterdam, The Netherlands; §California Northstate University College of Medicine, Elk Grove, United States; #The Lancet Oncology, London, United Kingdom; ¶Altius Institute for Biomedical Sciences, Seattle, United States; **BioSpyder Technologies, Carlsbad, United States

**Abstract** Human lung adenocarcinoma exhibits a propensity for de-differentiation, complicating diagnosis and treatment, and predicting poorer patient survival. In genetically engineered mouse models of lung cancer, expression of the BRAF[V600E] oncoprotein kinase initiates the growth of benign tumors retaining characteristics of their cell of origin, AT2 pneumocytes. Cooperating alterations that activate PI3'-lipid signaling promote progression of BRAF[V600E]-driven benign tumors to malignant adenocarcinoma. However, the mechanism(s) by which this cooperation occurs remains unclear. To address this, we generated mice carrying a conditional Braf[CAT] allele in which CRE-mediated recombination leads to co-expression of BRAF[V600E] and tdTomato. We demonstrate that co-expression of BRAF[V600E] and PIK3CA[H1047R] in AT2 pneumocytes leads to rapid cell de-differentiation, without decreased expression of the transcription factors NKX2-1, FOXA1, or FOXA2. Instead, we propose a novel role for PGC1α in maintaining AT2 pneumocyte identity. These findings provide insight into how these pathways may cooperate in the pathogenesis of human lung adenocarcinoma.
DOI: https://doi.org/10.7554/eLife.43668.001

## Introduction

Non-small cell lung cancer (NSCLC) is the leading cause of cancer-related death, with lung adenocarcinoma (LUAD) being the most common NSCLC subtype (Siegel et al., 2016). Due to the morbidity and mortality associated with LUAD, there is an urgent need to better characterize how key genetic drivers contribute to the pathogenesis of this disease. To that end, since the original discovery of KRAS mutations in human lung cancer cells (Capon et al., 1983), it has emerged that ~75% of LUADs display mutational activation of key components of receptor tyrosine kinase (RTK) signaling that, in turn, promote activation of RAS and its key downstream effectors: the RAF→MEK→ERK→MAP kinase (MAPK) and the PI3'-lipid pathways (Cancer Genome Atlas Research Network, 2014; Heist and Engelman, 2012). Moreover, mutational activation of RTKs or downstream signaling proteins (e.g. EGFR/ERBB1, ALK, ROS1, NTRK, BRAF) serve as predictive biomarkers for the clinical deployment of FDA-approved inhibitors of these oncoprotein kinases for the treatment of genetically-defined subsets of lung cancer (Drilon et al., 2018; Hyman et al., 2015; Rosell et al., 2012; Scagliotti et al., 2010; Shaw et al., 2013; Shaw et al., 2014).

**eLife digest** Cancers appear when changes in the genetic information of a cell, also called mutations, allow it to multiply uncontrollably. The disease we know as "lung cancer" kills more people than any other cancer, but this term actually refers to different types of tumors that appear because of various mutations that happen in different kinds of lung cells.

To complicate matters further, as lung cancer cells become more aggressive, they can stop appearing and behaving like the type of lung cell they came from. Yet, knowing the exact origin of the cancer is key, since it determines which treatment will work best to stop the disease in its tracks.

Despite these differences, many lung cancer cells contain mutations that over-activate two molecular cascades called the MAP kinase and the PI3'-kinase pathways. Under normal conditions, these signaling pathways relay external messages to the inside of the cell, where they help cells multiply. Two separate mutations can respectively over-stimulate either the MAP kinase or the PI3'-kinase pathway, but it was unclear how these could work together to start and maintain aggressive lung tumors. Another unanswered question was how these cancer cells lose the characteristics of the healthy cells they came from.

To address these issues, van Veen et al. genetically engineered mice that carry a mutation which activates the MAP kinase pathway. The lung cells with this genetic change also made a red fluorescent protein that marked cancer cells, so that these could be separated from the rest of the lung and analyzed.

This revealed that cells with only the MAP kinase mutation turned into small and benign tumors that began in lung cells, known as "type 2" cells. The PI3'-kinase mutation alone could not even start a tumor. However, together the mutations made tumors much more aggressive. Cells that carried both mutations also stopped producing proteins normally made by type 2 cells, therefore causing the cells to lose their original identity.

The mice created by van Veen et al. could help to understand how lung cancers develop in these animals and also in human lung cancer patients. Ultimately, this information could be used to design new cancer treatments, especially since both the MAP kinase and PI3'-kinase pathways contain many proteins that can be targeted with drugs.

DOI: https://doi.org/10.7554/eLife.43668.002

Mutational activation of BRAF occurs in ~8% of LUAD, with the most common single mutation ($BRAF^{T1799A}$) encoding the $BRAF^{V600E}$ oncoprotein kinase (*Cancer Genome Atlas Research Network, 2014*). To model $BRAF^{V600E}$ driven cancers, we previously described $Braf^{CA}$ mice carrying a CRE-activated allele of *Braf* that expresses normal BRAF prior to CRE-mediated recombination, after which $BRAF^{V637E}$ (orthologous to human $BRAF^{V600E}$ and for simplicity henceforth referred to as $BRAF^{V600E}$), is expressed from the endogenous chromosomal locus (*Dankort et al., 2007*). This mouse has proven useful in modeling many cancer types in which $BRAF^{V600E}$ is implicated as a driver oncoprotein (*Charles et al., 2011*; *Dankort et al., 2009*; *Sakamoto et al., 2017*; *Trejo et al., 2013*; *Wang et al., 2012*). Taken together, these studies indicate that *BRAF* mutation serves as a foundational initiating event for tumorigenesis in many target tissues. However, the progression of benign tumors initiated by $BRAF^{V600E}$ expression into malignant cancer invariably requires additional events such as silencing of tumor suppressors (e.g. INK4A-ARF, TP53, PTEN, CDX2) or activation of cooperating oncogenes (PIK3CA, CTNNB1, c-MYC) (*Charles et al., 2014*; *Dankort et al., 2007*; *Dankort et al., 2009*; *Huillard et al., 2012*; *Juan et al., 2014*; *Sakamoto et al., 2017*; *Trejo et al., 2013*; *Tsao et al., 2004*; *Yu et al., 2009*).

In mouse models of lung carcinogenesis, there are key similarities between the early stages of tumorigenesis observed in response to expression of either the $KRAS^{G12D}$ or $BRAF^{V600E}$ oncoproteins (*Dankort et al., 2007*; *Trejo et al., 2012*). While tumors initiated by $BRAF^{V600E}$ remain as benign adenomas with certain features of senescence (*Dankort et al., 2007*; *Jackson et al., 2001*), a proportion of $KRAS^{G12D}$ initiated lung tumors progress to frank adenocarcinomas within six months, most likely due to the ability of $KRAS^{G12D}$ to activate the PI3'-kinase signaling pathway (*Castellano et al., 2013*; *Murillo et al., 2018*; *Rodriguez-Viciana et al., 1994*; *Vivanco and Sawyers, 2002*; *Yuan and Cantley, 2008*). Consistent with this hypothesis, co-expression of $BRAF^{V600E}$

and PIK3CA[H1047R], a mutationally-activated form of PI3'-kinase-α (PI3Kα), in AT2 pneumocytes leads to rapid growth of lung tumors, many of which display progression to frank malignancy bearing various hallmarks of the cognate human disease (*Kinross et al., 2012*; *Trejo et al., 2013*). Thus, these genetically manipulated mice provide a unique opportunity to genetically and biochemically separate and analyze the effects of activation of these two critical downstream arms of RTK→RAS signaling individually or in combination.

Whereas the original *Braf*[CA] mouse allowed insights into cancer initiation, progression and therapy, there remain many questions that this mouse is inadequately configured to address. For example, it is not trivial to identify and isolate pure populations of tumor cells without significant stromal contamination, particularly in contexts when tumor cells are rare, such as in the earliest stages of tumorigenesis, or in the context of minimal residual disease following pathway-targeted inhibition of BRAF[V600E] signaling (*Dankort et al., 2007*). To address these issues we and others have used mice carrying CRE-activated alleles that express fluorescent proteins, such as the *mT-mG* allele in which the activity of CRE recombinase silences the expression of tdTomato and elicits expression of EGFP (*Muzumdar et al., 2007*). However, this approach is confounded by the observation that not all cells expressing the desired oncoprotein also express EGFP and *vice versa*.

In order to unequivocally identify BRAF[V600E] expressing cells we have generated *Braf*[CAT] mice carrying a new CRE-activated *Braf* allele. Like the original *Braf*[CA] allele, *Braf*[CAT] encodes normal BRAF prior to CRE-mediated recombination, after which the recombined allele expresses a bicistronic *Braf*[T1910A]-*P2A-tdTomato* mRNA encoding both BRAF[V600E] and the red fluorophore tdTomato. Moreover, here we report the use of *Braf*[CAT] mice to explore the cooperation of oncogenic BRAF[V600E] and PI3Kα[H1047R] in lung carcinogenesis in greater mechanistic detail. In brief, BRAF[V600E]-driven lung tumors maintain expression of markers of AT2 identity, including the known regulators of AT2 identity, NKX2-1, FOXA1, and FOXA2 (*Bruno et al., 1995*; *Camolotto et al., 2018*; *DeFelice et al., 2003*; *Hamvas et al., 2013*; *Lazzaro et al., 1991*; *Minoo et al., 1995*; *Snyder et al., 2013*; *Stahlman et al., 1996*; *Winslow et al., 2011*; *Yuan et al., 2000*). By contrast, co-expression of BRAF[V600E] and PI3Kα[H1047R] leads to development of lung tumors that show variable and widespread loss of expression of markers of AT2 pneumocyte terminal differentiation including the well-characterized surfactant proteins, SFTPA, SFTPB, and SFTPC. Notably reduced expression of AT2 markers begins early in tumor development and occurs despite sustained NKX2-1, FOXA1, and FOXA2 expression in tumor cells. Hence, these data shed light on the mechanisms by which pathways that cooperate in lung tumorigenesis also cooperate to influence the differentiation state of tumor cells. Indeed, these findings bear similarity to observations in human lung adenocarcinomas in which poorly differentiated and metastatic cancers often show loss of expression of functional markers of lung identity despite maintaining expression of NKX2-1 (*Yatabe et al., 2002*). Consequently, our results may shed light on our understanding of human lung cancer progression and how normal lung epithelial cells may lose their differentiation status following activation of cooperating oncogenic pathways.

## Results

### Generation of *Braf*[CAT] mice

To generate a reporter of BRAF[V600E] oncoprotein expression, we linked its expression to the expression of the red fluorophore, tdTomato. To accomplish this, we made use of the design of the original *Braf*[CA] allele, in which the modified exon 18 and the remainder of the *Braf* allele is not transcribed prior to the action of CRE recombinase (*Dankort et al., 2007*) due to the insertion of a triple polyadenylation/mRNA transcription termination signal from SV40 (*Figure 1A*) (*Srinivas et al., 2001*). Consequently, we designed a targeting vector containing the final coding exon (22) of mouse *Braf* in which the stop codon was removed, followed by sequences encoding: 1. an in-frame glycine-serine-glycine-P2A self-cleaving peptide; 2. sequences encoding a membrane-tethered tdTomato-CAAX protein and; 3. a PGK-PURO selectable marker flanked by Frt sites for subsequent removal by FLP recombinase. Following electroporation of this construct into 2H1 *Braf*[CA/+] ES cells, from which the original *Braf*[CA] mice were generated, 288 puromycin resistant clones were selected and screened by PCR for homologous recombination of the construct into the distal end of the *Braf*[CA] allele (*Dankort et al., 2007*). However, because the targeted ES cells are heterozygous for both

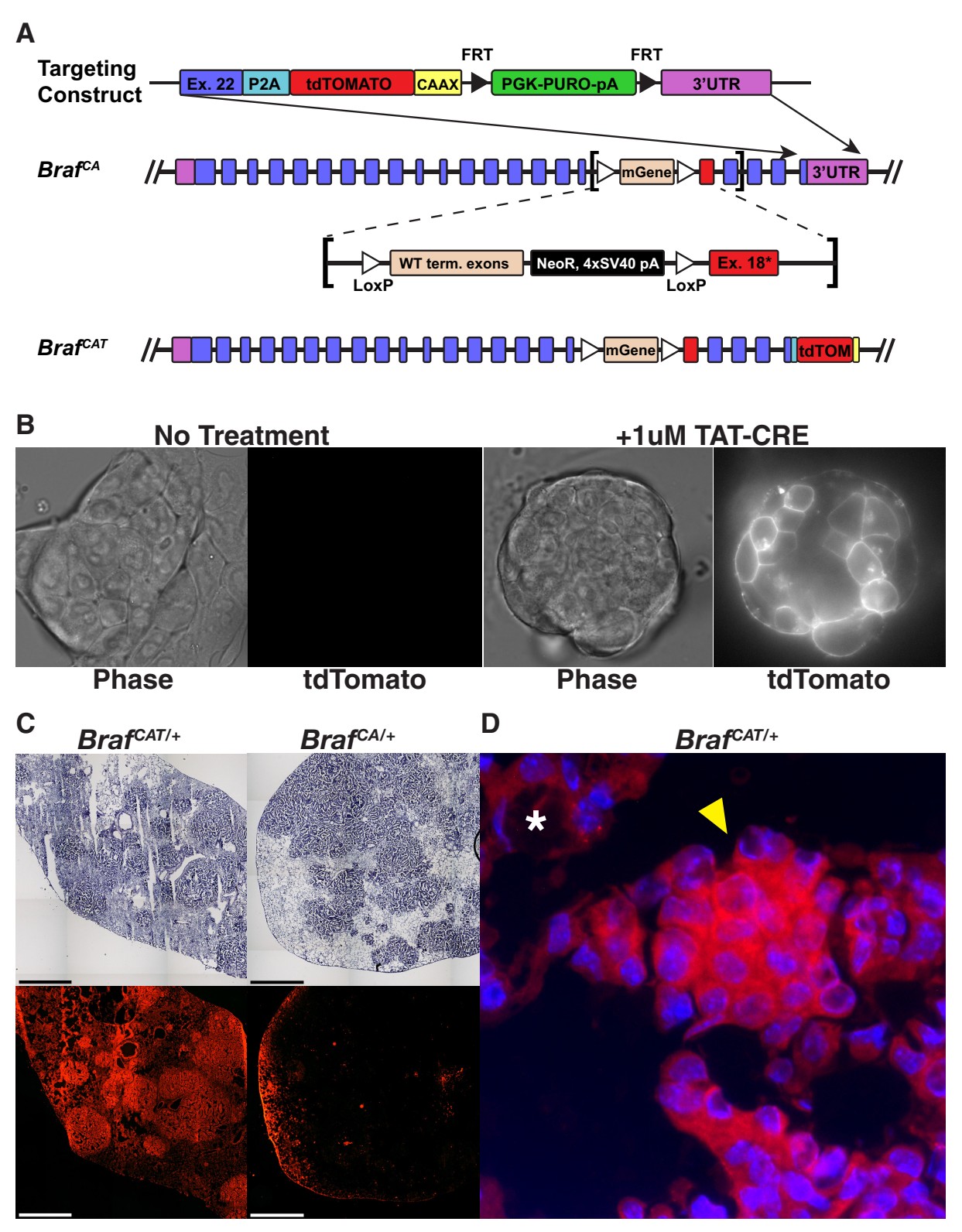

**Figure 1.** Engineering and validation of a novel genetically engineered mouse model of BRAFV600E driven cancer. (**A**) The *Braf^CAT* mouse builds upon the utility of the *Braf^CA* mouse by tying expression of the oncogenic form of BRAF to expression of the red fluorescent protein, tdTomato. (**B**) Targeted *Braf^CAT* ES cells display membrane associated red fluorescence only after the addition of TAT-CRE. (**C**) Comparison of Ad-CMV-CRE initiated lung
*Figure 1 continued on next page*

*Figure 1 continued*

tumor formation and fluorescence in frozen sections from lungs of *Braf*[CAT/+] and *Braf*[CA/+] animals. (D) Lung adenoma found in a *Braf*[CAT] animal showing fluorescence in the tumor (arrowhead) and not in the lung parenchyma (asterisk).

DOI: https://doi.org/10.7554/eLife.43668.003

The following figure supplement is available for figure 1:

**Figure supplement 1.** Engineering and validation of a novel genetically engineered mouse model of BRAFV600E driven cancer.

DOI: https://doi.org/10.7554/eLife.43668.004

normal *Braf* and the genetically manipulated *Braf*[CA] allele, and because there was no way to direct homologous recombination of the targeting vector to the previously targeted *Braf*[CA] allele, we expected to target both homologues. Because BRAF is expressed in ES cells, we reasoned that modification of the normal allele would lead to ES cells with constitutive tdTomato-CAAX expression. Indeed, ~50% of PCR positive ES cell clones displayed constitutive membrane associated red-fluorescence and were used to generate *Braf*[TOM] mice, in which tdTomato serves as a marker for any cells expressing normal BRAF (*van Veen et al., 2016*). By contrast, homologous recombination of the targeting vector into the *Braf*[CA] allele should give rise to ES cells that do not express tdTomato due to the strong transcriptional termination signal. However, upon the addition of a cell permeable TAT-CRE protein to these cells, they should initiate the expression of both BRAF[V600E] and tdTomato (*Figure 1B*). Hence, this in vitro strategy allowed us to both identify appropriately targeted *Braf*[CAT/+] ES cells and also indicated the appropriate functioning of the *Braf*[CAT] allele in response to CRE-mediated recombination prior to the generation of mice.

*Braf*[CAT] mice were generated from an appropriately targeted ES cell clone (1E6). To compare and contrast lung tumorigenesis following CRE-mediated recombination of the *Braf*[CAT] versus the original *Braf*[CA] allele, mice of the appropriate genotype were infected with $10^7$ pfu of Ad-CMV-CRE and analyzed at 8 weeks post-initiation (p.i.). We observed similar lung tumor formation in *Braf*[CAT] versus *Braf*[CA] mice (*Figure 1C*) with the only discernable difference being the red fluorescence of lung tumors arising in the *Braf*[CAT] mice (*Figure 1C,D* and *Figure 1—figure supplement 1A*). In embryonic fibroblasts (MEFs) derived from *Braf*[CAT] mice, tdTomato fluorescence was detected by flow cytometry within 24 hr after expression of CRE and plateaued by 96 hr (*Figure 1—figure supplement 1B*). Following CRE-mediated recombination of *Braf*[CAT] in both MEFs and mouse lung cells (*Figure 2B*), the total amount of fluorescence from the *Braf*[CAT] allele was modest, likely due to being driven by the endogenous *Braf* promoter. However, tdTomato expressing cells were readily differentiated from autofluorescence by using a channel with no fluorophore (FITC) as a marker of autofluorescence, as has been previously described (*Dane et al., 2006*). Together, these data indicate that the *Braf*[CAT] allele functions analogously to the *Braf*[CA] allele for the development of benign lung tumors following CRE-mediated initiation of BRAF[V600E] expression, and that cells expressing the BRAF[V600E]-P2A-tdTomato mRNA are readily identified by flow cytometry within a short time frame.

## Messenger RNA expression profiles of BRAF[V600E]/PI3Kα[H1047R]-driven lung tumors display diminished expression of AT2 pneumocyte specific genes

To address mechanism(s) of cooperation between BRAF[V600E] and PI3'-lipid signaling in lung cancer progression, lung tumorigenesis was initiated in *Braf*[CAT] or *Braf*[CAT]; *Pik3ca*[lat-H1047R] (*Pik3ca*[HR] hereafter) mice (*Figure 2A*) and analyzed at 2, 6 or 12 weeks p.i. (*Figure 2B*, for detailed gating strategy see *Figure 2—figure supplement 1A*). Importantly, to initiate oncoprotein expression solely in AT2 pneumocytes, we utilized Ad5-SpC-CRE, which restricts expression of CRE recombinase to *Sftpc* expressing cells (*Sutherland et al., 2014*). tdTomato expressing tumor cells were detectable by flow cytometry in both *Braf*[CAT] and *Braf*[CAT]; *Pik3ca*[HR] mice as early as two weeks p.i. (*Figure 2B*). To identify alterations in mRNA expression that might explain how PI3Kα[H1047R] promotes malignant transformation of lung tumors initiated by BRAF[V600E], we performed RNA-Seq analysis of flow sorted tdTomato[+] lung tumor cells driven either by BRAF[V600E] alone or the combination of BRAF[V600E] plus PI3Kα[H1047R]. To gain a broad view of pathways and processes differing in these two tumor genotypes, we used Gene Set Enrichment Analyses (GSEA) (*Figure 2C*) on, samples from all time points (for GSEA analyses separated by week see *Figure 2—figure supplement 1B*). Comparing 'hallmark'

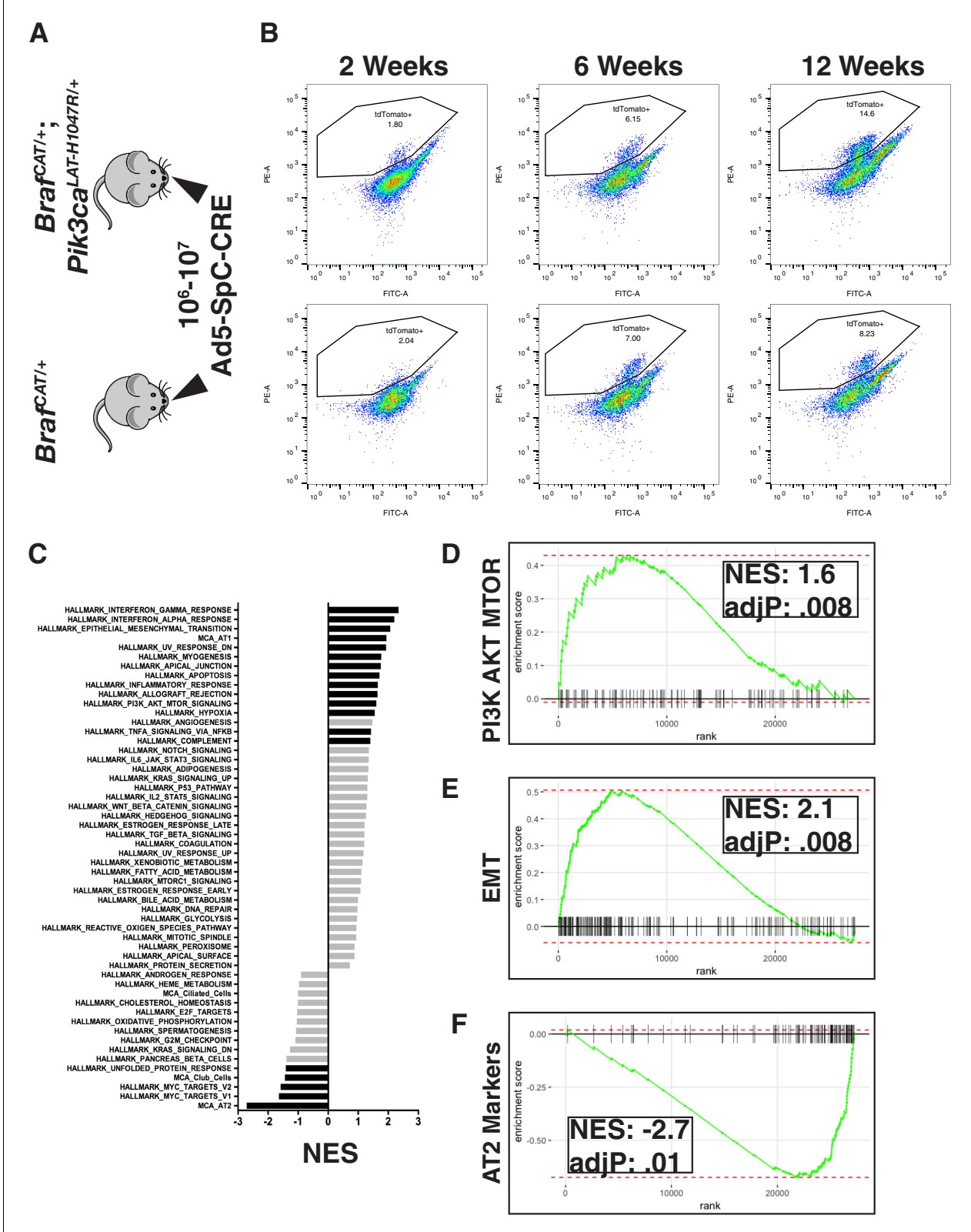

**Figure 2.** Examining global gene expression changes caused by combination of PI3K and MAPK activation using RNA-SEQ. (**A**) Tumors were adenovirally induced in cohorts of *Braf^CAT/+* and *Braf^CAT/+;Pik3ca^LAT-H1047R/+* mice with AT2 specific CRE expression. Mice harvested at 2 or 6 weeks were induced with $10^7$ PFU of Ad5-SpC-CRE, whereas mice harvested at 12 weeks were induced with $10^6$ PFU of Ad5-SpC-CRE. (**B**) Tumor cells were harvested from each genotype at 2 weeks, 6 weeks, and 12 weeks post tumor induction via tissue dissociation and FACS. (**C**) GSEA analyses profiling
*Figure 2 continued on next page*

*Figure 2 continued*

cells sorted from BRAF$^{V600E}$/PI3K$\alpha^{H1047R}$ driven tumor bearing mice compared to BRAF$^{V600E}$ driven tumor bearing mice, showing all 'Hallmark' gene sets along with gene sets constructed from the most specific markers of the cell types of the distal lung epithelium. Black bars indicate adjP <.05, gray bars indicate Benjamini-Hochberg corrected enrichment statistic adjP $\geq$. 05. Here all time points combined within genotype. (D) GSEA mountain plot showing broad activation of PI3K signaling in BRAF$^{V600E}$/PI3K$\alpha^{H1047R}$ driven tumor bearing mice; adjP is Benjamini-Hochberg corrected enrichment statistic. (E) GSEA mountain plot showing broad activation of EMT in BRAF$^{V600E}$/PI3K$\alpha^{H1047R}$ driven tumor bearing mice; adjP is Benjamini-Hochberg corrected enrichment statistic. (F) GSEA mountain plot showing widespread loss of AT2 identity in BRAF$^{V600E}$/PI3K$\alpha^{H1047R}$ driven tumor bearing mice; adjP is Benjamini-Hochberg corrected enrichment statistic.

DOI: https://doi.org/10.7554/eLife.43668.005

The following source data, source code and figure supplements are available for figure 2:

**Source code 1.** R script to perform gene set enrichment analysis on *Figure 2—source data 1–2*, as well as plot these results.
DOI: https://doi.org/10.7554/eLife.43668.007
**Source data 1.** DEseq2 output of differentially expressed genes comparing BRAF$^{V600E}$/PI3K$\alpha^{H1047R}$ and BRAF$^{V600E}$ driven tumors – all weeks pooled.
DOI: https://doi.org/10.7554/eLife.43668.008
**Source data 2.** DEseq2 output of differentially expressed genes comparing BRAF$^{V600E}$/PI3K$\alpha^{H1047R}$ and BRAF$^{V600E}$ driven tumors – weeks separated.
DOI: https://doi.org/10.7554/eLife.43668.009
**Figure supplement 1.** Examining global gene expression changes caused by combination of PI3K and MAPK activation using RNA-SEQ.
DOI: https://doi.org/10.7554/eLife.43668.006

gene sets (Broad Institute MSigDB: Hallmarks), and consistent with the engineered characteristics of the lung tumor cells, GSEA revealed that PI3K→AKT→MTOR signaling (*Figure 2C and D*) and epithelial→mesenchymal transition (*Figure 2C and E*) related genes were significantly elevated in BRAF$^{V600E}$/PI3K$\alpha^{H1047R}$-driven lung tumors compared to BRAF$^{V600E}$-driven tumors. To examine differentiation state we constructed gene sets comprised of the 100 most specific described mRNA markers of the different cell types expressed in the distal lung epithelium (*Han et al., 2018*; *Treutlein et al., 2014*) namely alveolar type 1 (AT1) and type 2 (AT2) pneumocytes, as well as club and ciliated cells, hereafter referred to as AT1-100, AT2-100, club-100, and ciliated-100, respectively. Despite these gene sets representing related cell types of the distal lung epithelium, there is only modest overlap between their members, ranging from 2 to 12 of the 100 genes. GSEA using these gene sets demonstrated that, compared to BRAF$^{V600E}$ expressing tumor cells, BRAF$^{V600E}$/PI3K$\alpha^{H1047R}$ expressing tumor cells showed a significant decrease (adj. p=0.01) in expression of transcripts associated with AT2 cell identity (*Figure 2C and F*). This effect encompassed nearly all AT2 markers including the classical markers, *Sftpa/b/c*, and newly described markers detected in many different gene classes including *Lcn2* (lipid transporter), *Bex2* (transcription factor), and *Dlk1* (encoding a non-canonical NOTCH ligand). Together these data suggest that PI3K$\alpha^{H1047R}$ signaling promotes reduced expression of markers of AT2 differentiation. Notably, when examined in parallel with 50 hallmark gene sets and markers of other cell types in the distal lung epithelium, the loss of expression of AT2 mRNAs was the strongest effect observed in association with PI3K$\alpha^{H1047R}$ expression (NES: −2.71, *Figure 2C and F*). Decreased AT2 marker expression was statistically significant as early as 2 weeks (*Figure 3A* and *Figure 2—figure supplement 1B*) and was also observed at 6 weeks (*Figure 3B* and *Figure 2—figure supplement 1B*) and 12 weeks (*Figure 3C* and *Figure 2—figure supplement 1B*) p.i. Hence the effects of PI3K$\alpha^{H1047R}$ on expression of AT2 differentiation markers initiates more rapidly than has been reported in tumors driven by KRAS$^{G12D}$ (*Desai et al., 2014*), KRAS$^{G12D}$/TP53$^{Null}$ (*Winslow et al., 2011*), KRAS$^{G12D}$/CTNNB1$^{\Delta ex3}$ (*Pacheco-Pinedo et al., 2011*), BRAF$^{V600E}$/TP53$^{Null}$ (*Garnett et al., 2017*; *Shai et al., 2015*), or BRAF$^{V600E}$/CTNNB1$^{\Delta ex3}$ (*Juan et al., 2014*). In addition to reduced AT2 mRNA marker expression, PI3K$\alpha^{H1047R}$ expression elicited a marked increase in AT1 marker expression when comparing all time points (*Figure 2C*). By contrast to the changes observed in AT2 marker expression, this change was most significant at earlier time points and was no longer observed to be significant by 12 weeks p.i. (*Figure 2—figure supplement 1B*). Together, these results suggest a capability of PI3'-lipid signaling to influence AT2 pneumocyte identity.

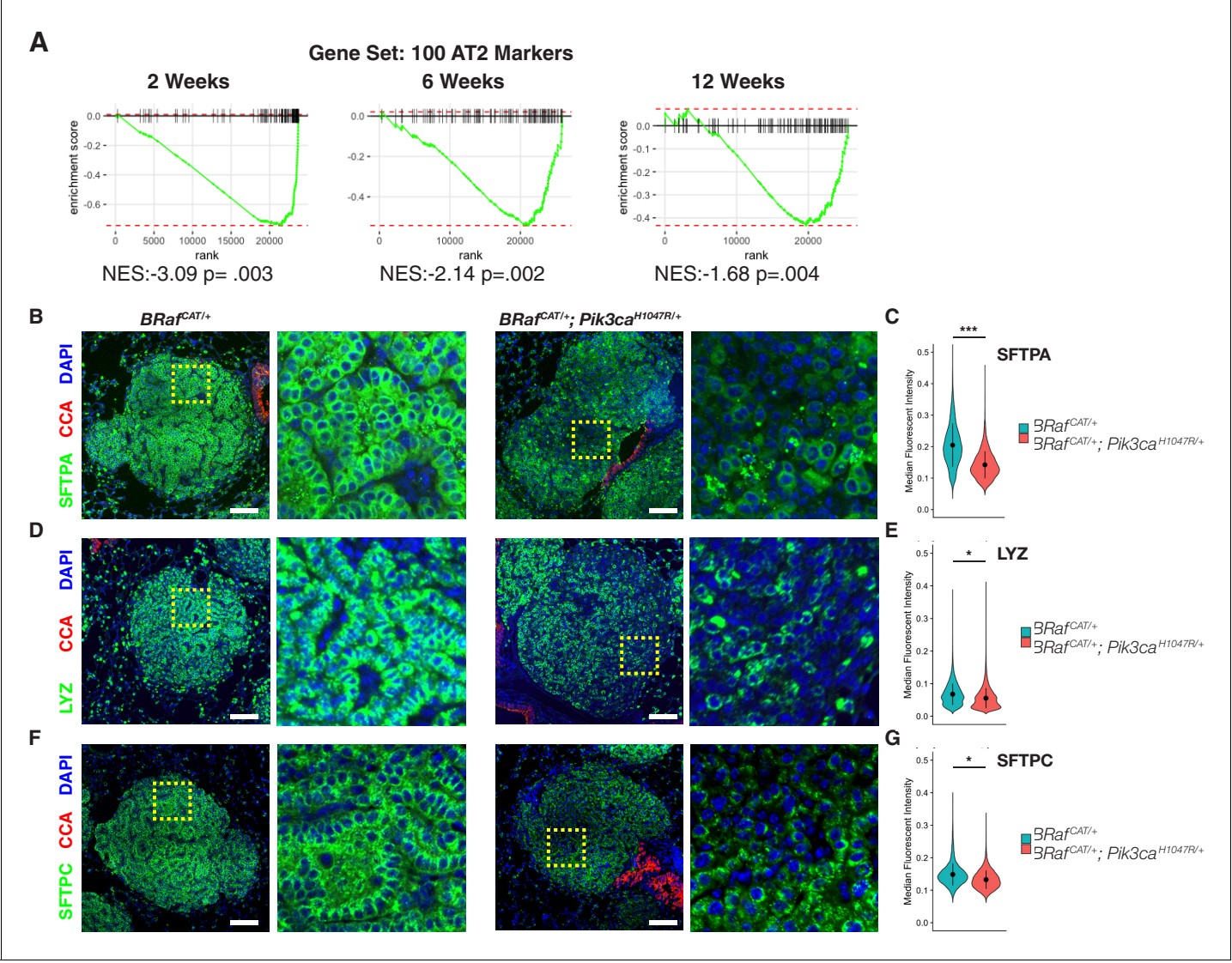

**Figure 3.** BRAF[V600E]/PI3Kα[H1047R] Driven Tumors Display Widespread Heterogeneous Loss of AT2 Marker Expression, Whereas BRAF[V600E] Driven Tumors Maintain AT2 marker expression. (**A**) GSEA mountain plots comparing gene expression profiles of cells sorted from *Braf[CAT/+];Pik3ca[LAT-H1047R/+]* mice to those from *Braf[CAT/+]* mice, using 100 specific markers of AT2 identity as the gene set. Analyses show comparison of tumors from the two genotypes at each time point analyzed; P value is enrichment statistic. (**B**) Tumors found in *Braf[CAT/+];Pik3ca[LAT-H1047R/+]* mice have widespread variegated loss of expression of the functional AT2 marker, SFTPA, compared to tumors found in *Braf[CAT/+]* mice. Expression of CCA is seen in airways but not in tumors of either genotype. Dashed boxes highlight areas of increased magnification. Scale bars = 100 um. (**C**) CellProfiler based quantitation of SFTPA immunofluorescence. Wilcoxon rank sum p value = 0.00013 (**D**) Tumors found in *Braf[CAT/+];Pik3ca[LAT-H1047R/+]* mice have widespread loss of expression of the functional AT2 marker, LYZ, compared to tumors found in *Braf[CAT/+]* mice. (**E**) CellProfiler based quantitation of LYZ immunofluorescence. Wilcoxon rank sum p value = 0.02224 (**F**) Tumors found in *Braf[CAT/+];Pik3ca[LAT-H1047R/+]* mice have widespread loss of expression of the functional AT2 marker, SFTPC, compared to tumors found in *Braf[CAT/+]* mice. (**G**) CellProfiler based quantitation of SFTPC immunofluorescence. Wilcoxon rank sum p value = 0.02323.

DOI: https://doi.org/10.7554/eLife.43668.010

The following source data, source code and figure supplements are available for figure 3:

**Source code 1.** R script to perform gene set enrichment analyses on *Figure 2—source data 2*, as well as plot these results.
DOI: https://doi.org/10.7554/eLife.43668.012
**Source code 2.** R script to perform statistics on *Figure 3—source data 1–3*, as well as plot these results.
DOI: https://doi.org/10.7554/eLife.43668.013
**Source code 3.** Cellprofiler pipeline to quantify raw images, producing *Figure 3—source data 1–3*.
DOI: https://doi.org/10.7554/eLife.43668.014
**Source data 1.** Cellprofiler output quantifying SFTPA immunofluorescence in BRAF[V600E]/PI3Kα[H1047R] and BRAF[V600E] driven tumors.
*Figure 3 continued on next page*

*Figure 3 continued*

DOI: https://doi.org/10.7554/eLife.43668.015

**Source data 2.** Cellprofiler output quantifying LYZ immunofluorescence in BRAF$^{V600E}$/PI3K$\alpha^{H1047R}$ and BRAF$^{V600E}$ driven tumors.

DOI: https://doi.org/10.7554/eLife.43668.016

**Source data 3.** Cellprofiler output quantifying SFTPC immunofluorescence in BRAF$^{V600E}$/PI3K$\alpha^{H1047R}$ and BRAF$^{V600E}$ driven tumors.

DOI: https://doi.org/10.7554/eLife.43668.017

**Figure supplement 1.** Cellprofiler based immunofluorescence quantification strategy.

DOI: https://doi.org/10.7554/eLife.43668.011

## PI3K$\alpha^{H1047R}$ cooperates with BRAF$^{V600E}$ expression to promote loss of expression of lung tumor differentiation markers

Consistent with the mRNA expression data, *Ad5-Sftpc-CRE* initiated BRAF$^{V600E}$/ PI3K$\alpha^{H1047R}$-driven lung tumors displayed decreased expression of surfactant proteins A and C (SFTPA and SFTPC) and Lysozyme, all of which are AT2 pneumocyte markers (*Figure 3B,D and F*). We next built a pipeline in CellProfiler, which allows for quantification of immunofluorescence images with single cell resolution in thousands of cells algorithmically (*Figure 3—figure supplement 1A–H*). By this means we noted significant reductions of SFTPA, SFTPC, and LYZ expression at 12 weeks p.i. (*Figure 3C, E and G*. Wilcoxon p=0.0001,. 02,. 02 respectively, data from *Figure 3—source data 1*, *2* and *3*).

It has previously been suggested that a population of cells at the bronchioalveolar junction co-expressing SFTPC and club cell antigen (CCA) has properties of <u>b</u>ronchio-<u>a</u>lveolar <u>s</u>tem <u>c</u>ells (BASCs) (*Kim et al., 2005*), and also that ERK1/2 signaling tone allows for expansion of club cell derived tumors (*Cicchini et al., 2017*). Thus, a hypothesis that might explain our observations is that expression of PI3K$\alpha^{H1047R}$ allows for enhanced expansion of BASC-derived tumors, which express lower levels of AT2 marker genes. We reject this hypothesis based on three lines of evidence. First, there did not appear to be two classes of BRAF$^{V600E}$/PIK3CA$^{H1047R}$-induced tumors with respect to AT2 marker expression. Instead, within each tumor we observed variegated loss of AT2 marker expression (*Figure 3B,D,F* insets). Second, to test whether CCA/SFTPC double positive cells might be the source of emergence of a separate tumor type, we co-stained our AT2 marker panel with antisera to detect expression of club cell antigen (CCA). While CCA staining was readily detectable in airways, it was not observed throughout the BRAF$^{V600E}$/PI3K$\alpha^{H1047R}$-driven tumors, including those cells that have reduced AT2 marker expression (*Figure 3B,D and F*). Finally, we observed tumors arising predominantly within alveolar spaces, a pattern characteristic of AT2-derived tumors, whereas club cell derived tumors tend to arise predominantly at bronchioalveolar duct junctions (*Cicchini et al., 2017*). Taken together, these data suggest that BRAF$^{V600E}$/PI3K$\alpha^{H1047R}$-driven lung tumors arise from AT2 cells but rapidly lose their differentiated identity under the influence of PI3'-lipid signaling.

## Cooperative signaling between PI3K$\alpha^{H1047R}$ and BRAF$^{V600E}$ promotes de-differentiation of AT2 cells despite expression of NKX2-1

Extensive research demonstrates that the NKX2-1 and FOXA1/2 transcription factors pattern and maintain the differentiated identity of normal lung cells (*Bruno et al., 1995*; *Camolotto et al., 2018*; *DeFelice et al., 2003*; *Hamvas et al., 2013*; *Lazzaro et al., 1991*; *Minoo et al., 1995*; *Snyder et al., 2013*; *Stahlman et al., 1996*; *Yuan et al., 2000*). As these transcription factors also display decreased expression in some models of LUAD (*Juan et al., 2014*; *Snyder et al., 2013*; *Winslow et al., 2011*), we next examined if changes in bulk NKX2-1 or FOXA1/2 expression in BRAF$^{V600E}$/PIK3CA$^{H1047R}$-driven lung tumors might explain the observed alterations in AT2 marker gene expression. Analysis of RNA-Seq data showed no consistent decrease of NKX2-1 or FOXA1/2 mRNA expression in BRAF$^{V600E}$/PIK3CA$^{H1047R}$-driven lung tumors. As this was initially surprising, we sought to further verify that AT2 identity was lost independent of NKX2-1 expression levels or localization in BRAF$^{V600E}$/PIK3CA$^{H1047R}$-driven lung tumors. Dual color immunofluorescence staining demonstrated that the observed loss of AT2 identity is not associated with a decrease or change in nuclear localization of NKX2-1 at either 2 or 12 weeks p.i. (*Figure 4A–B*). Quantification of immunostaining supports this observation, with no significant alteration in nuclear NKX2-1 staining at 2 weeks, and only a slight but significant *increase* in nuclear NKX2-1 staining observed when comparing 12 week p.i. BRAF$^{V600E}$/PIK3CA$^{H1047R}$-driven tumors to paired BRAF$^{V600E}$-driven tumors

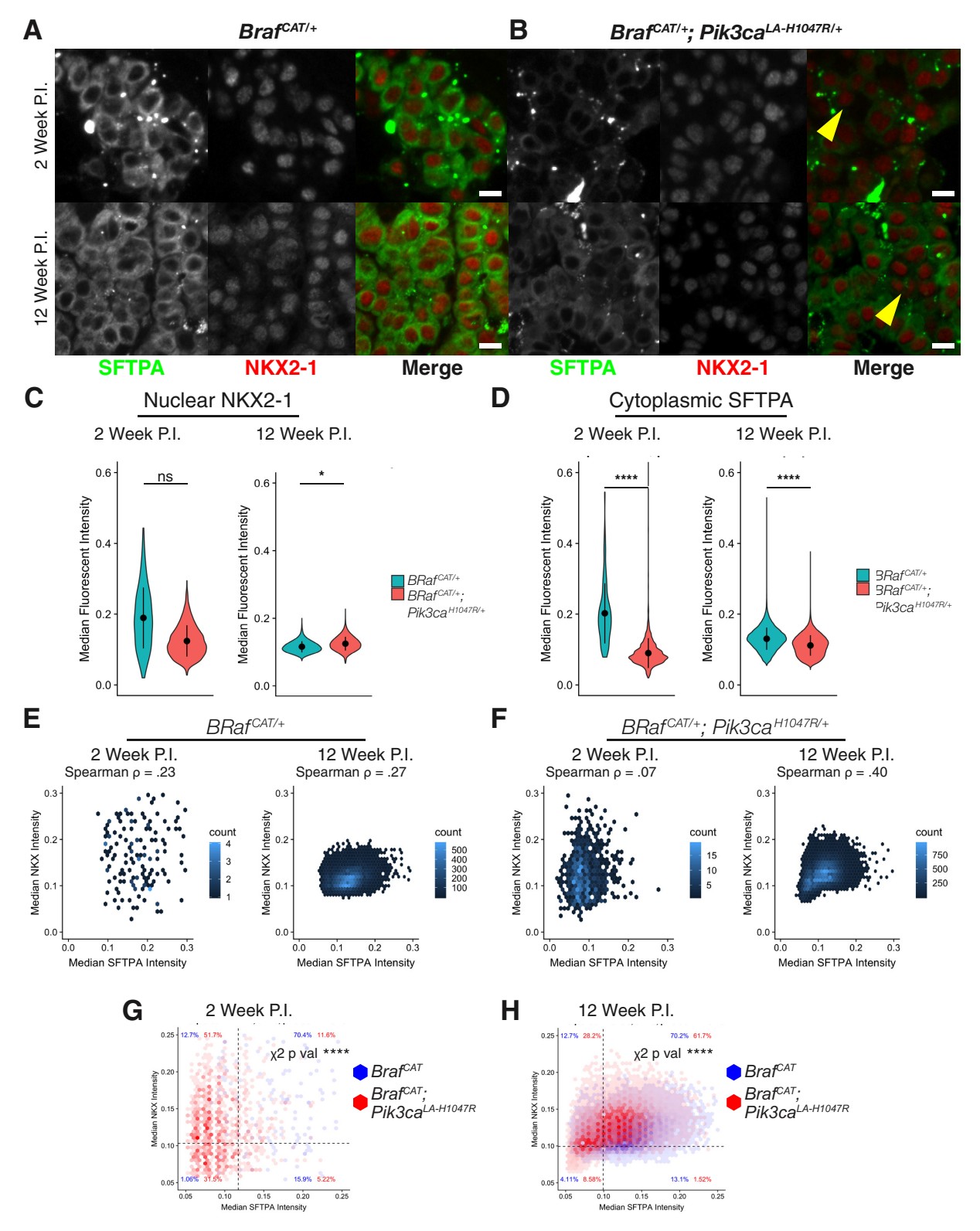

**Figure 4.** Expression levels and localization of lung lineage survival transcription factors are maintained in BRAF^V600E/PI3Kα^H1047R driven tumors, Including those cells which have lost expression of markers of AT2 identity. (A) BRAF^V600E driven hyperplasia and tumors display widespread expression of both SFTPA and nuclear localization of the lung lineage transcription factor, NKX2-1, at 2 and 12 weeks post initiation. Scale bar = 10 um. (B) BRAF^V600E/PI3Kα^H1047R driven hyperplasia and tumors show decreased SFTPA expression at 2 and 12 weeks post initiation. These tumors maintain

*Figure 4 continued on next page*

*Figure 4 continued*

broad nuclear expression of NKX2-1, including those cells with decreased SFTPA expression (yellow arrowheads). (C) Quantitation showing no significant difference in NKX2-1 immunoreactivity at 2 weeks post initiation, but a slight increase in nuclear NKX2-1 at 12 weeks post initiation. Wilcoxon rank sum p values = 0.2,. 02 respectively. (D) Significant reduction of SFTPA immunoreactivity seen in BRAF$^{V600E}$/PI3K$\alpha^{H1047R}$ driven hyperplasia and tumors at both 2 and 12 weeks post initiation. Wilcoxon rank sum p values = 5e-5, 4e-5 respectively. (E) Cytoplasmic SFTPA immunoreactivity plotted versus nuclear NKX2-1 immunoreactivity in BRAF$^{V600E}$ driven hyperplasia and tumors at 2 and 12 weeks post initiation. Similar association seen at both time points (Rho = 0.23,. 27 respectively). (F) Cytoplasmic SFTPA immunoreactivity plotted versus nuclear NKX2-1 immunoreactivity in BRAF$^{V600E}$/PI3K$\alpha^{H1047R}$ driven hyperplasia and tumors at 2 and 12 weeks post initiation. Relatively lower association seen at 2 weeks compared to 12 weeks (Rho = 0.07,. 40 respectively). (G) Overlay of BRAF$^{V600E}$/PI3K$\alpha^{H1047R}$ and BRAF$^{V600E}$ driven hyperplasia 2 weeks post initiation. Dashed line for each marker drawn at mean - one standard deviation of BRAF$^{V600E}$ driven tumors. BRAF$^{V600E}$/PI3K$\alpha^{H1047R}$ driven tumors show fewer SFTPA+, NKX2−1 + cells, most strongly accounted for by an increase in SFTPA-, NKX2−1 + cells. Chi square test associates genotype with distribution, p val <1e-5. (H) Overlay of BRAF$^{V600E}$/PI3K$\alpha^{H1047R}$ and BRAF$^{V600E}$ driven tumors 12 weeks post initiation. Dashed line for each marker drawn at mean - one standard deviation of BRAF$^{V600E}$ driven tumors. BRAF$^{V600E}$/PI3K$\alpha^{H1047R}$ driven tumors show fewer SFTPA+, NKX2−1 + cells, most strongly accounted for by an increase in SFTPA-, NKX2−1 + cells. Chi square test associates genotype with distribution, p val <1e-5.
DOI: https://doi.org/10.7554/eLife.43668.018

The following source data, source code and figure supplements are available for figure 4:

**Source code 1.** R script to perform statistics on *Figure 4—source data 1–2*, as well as plot these results.
DOI: https://doi.org/10.7554/eLife.43668.020
**Source code 2.** Cellprofiler pipeline to quantify raw images from BRAF$^{V600E}$/PI3K$\alpha^{H1047R}$ and BRAF$^{V600E}$ driven tumors, producing *Figure 4—source data 1*.
DOI: https://doi.org/10.7554/eLife.43668.021
**Source code 3.** Cellprofiler pipeline to quantify raw images from KRAS$^{G12D}$/PIK3CA$^{H1047R}$ and KRAS$^{G12D}$ driven tumors, producing *Figure 4—source data 2*.
DOI: https://doi.org/10.7554/eLife.43668.022
**Source data 1.** Cellprofiler output quantifying immunofluorescence of SFTPA and NKX2-1 in BRAF$^{V600E}$/PI3K$\alpha^{H1047R}$ and BRAF$^{V600E}$ driven tumors.
DOI: https://doi.org/10.7554/eLife.43668.023
**Source data 2.** Cellprofiler output quantifying immunofluorescence of SFTPA and NKX2-1 in KRAS$^{G12D}$/PIK3CA$^{H1047R}$ and KRAS$^{G12D}$ driven tumors.
DOI: https://doi.org/10.7554/eLife.43668.024
**Figure supplement 1.** Expression levels and localization of lung lineage survival transcription factors are maintained in BRAF$^{V600E}$/PI3K$\alpha^{H1047R}$ and KRAS$^{G12D}$/PI3K$\alpha^{H1047R}$ Driven Tumors, including those cells which have lost expression of markers of AT2 identity.
DOI: https://doi.org/10.7554/eLife.43668.019

(*Figure 4C*, data from *Figure 4—source data 1*). Quantification of the same tumors confirmed a significant decrease of SFTPA staining in BRAF$^{V600E}$/PIK3CA$^{H1047R}$-driven tumors first observed at 2 weeks and persisting at 12 weeks p.i. (*Figure 4D*). Since we performed our quantification with single cell resolution we were able to compare NKX2-1 and SFTPA immunofluorescence staining on a cell-by-cell basis (*Figure 4E–H*). BRAF$^{V600E}$-driven tumors show association of NKX2-1 and SFTPA staining consistent across time points (Spearman Rho = 0.23-.27, *Figure 4E*). BRAF$^{V600E}$/PIK3CA$^{H1047R}$-driven tumors by contrast initially show almost no association between levels of NKX2-1 and SFTPA (Spearman Rho = 0.07, *Figure 4F*), but by 12 weeks p.i. the association of these two markers has increased markedly (Spearman Rho = 0.40, *Figure 4F*). Because the reduction of SFTPA precedes the observed association of NKX2-1 and SFTPA, we conclude that other factors must explain the rapid reduction in SFTPA expression. Dividing tumor cells into classes representing NKX2−1 ± and SFTPA +/- (for definitions, see Materials and methods) shows significant effects (chi-squared p<0.001) of tumor genotype on co-expression of NKX2-1 and SFTPA (*Figure 4G–H*). At both 2 and 12 weeks p.i., the largest proportional increase driven by PIK3CA$^{H1047R}$ expression is seen in NKX2-1$^+$/SFTPA$^-$ tumor cells (*Figure 4G–H*), implying that at both early and late time points, decreased expression of NKX2-1 cannot explain the observed decrease in SFTPA expression. Similarly, neither the expression of FOXA1 (*Figure 4—figure supplement 1A*) nor FOXA2 (*Figure 4—figure supplement 1B*) at 12 weeks p.i. correlated with decreased SFTPA expression as assessed by immunostaining. Nor is the phosphorylation status of NKX2-1 at a critical ERK targeted residue (pS327) associated with loss of SFTPA expression (*Figure 4—figure supplement 1C*). Together these data suggest that the decreased expression of markers of AT2 cell differentiation that we observed upon co-activation of PI3'-lipid signaling in BRAF$^{V600E}$ driven tumors is largely independent of the expression levels of these known regulators of AT2 cell identity. We also note that at later time points,

repression of NKX2-1, FOXA1, or FOXA2 protein expression in scattered cells may serve to augment the dedifferentiation phenotype that we observed beginning at 2 weeks p.i.

To test the generality of our results, we examined KRAS$^{G12D}$ (*Figure 4—figure supplement 1D*) and KRAS$^{G12D}$/PIK3CA$^{H1047R}$-driven lung tumors (*Figure 4—figure supplement 1E*) at 16 weeks p.i., a time at which mutationally-activated PI3Kα$^{H1047R}$ strongly enhances KRAS$^{G12D}$-driven lung tumorigenesis (*Green et al., 2015*). Indeed, in this model, we observed a similar and significant decrease in SFTPA expression comparing KRAS$^{G12D}$/PIK3CA$^{H1047R}$- to KRAS$^{G12D}$-driven tumors (*Figure 4—figure supplement 1D,E,G*, data from *Figure 4—source data 2*), which did not correspond to reduced NKX2-1 expression (*Figure 4—figure supplement 1E,F*). Intriguingly, although mutationally-activated KRAS$^{G12D}$ is reported to activate PI3'-lipid signaling in lung tumors (*Castellano et al., 2013*; *Engelman et al., 2008*; *Gupta et al., 2007b*; *Molina-Arcas et al., 2013*; *Murillo et al., 2018*; *Rodriguez-Viciana et al., 1994*), we did not observe extensive loss of SFTPA immunoreactivity in these tumors (*Figure 4—figure supplement 1D*). Both KRAS$^{G12D}$/PIK3CA$^{H1047R}$-driven tumors and KRAS$^{G12D}$-driven tumors showed similarly modest association of SFTPA and NKX2-1 expression (*Figure 4—figure supplement 1H–I*). Taken together these data suggest that either there exist additional factors that regulate AT2 pneumocyte identity independently of the NKX2-1/FOXA1/FOXA2 regulatory axis, or that these well-known regulators of pneumocyte identity require the presence of one or more additional factor(s) for their transcriptional function.

To explore the increase in AT1 marker expression observed in BRAF$^{V600E}$/PI3Kα$^{H1047R}$-driven tumors in more detail, we immunostained both early and late tumors for the expression of the AT1 marker AQP5. At 2 weeks p.i. we observed a striking increase in AQP5 expression when comparing BRAF$^{V600E}$/PI3Kα$^{H1047R}$-driven tumors to BRAF$^{V600E}$-driven tumors (*Figure 5A and B*, data from *Figure 5—source data 1*). Interestingly, at 12 weeks p.i. the difference in AQP5 immunoreactivity is no longer significant between BRAF$^{V600E}$/PI3Kα$^{H1047R}$ and BRAF$^{V600E}$-driven tumors, mirroring our mRNA expression profiling results (*Figure 2* and *Figure 2—figure supplement 1b*). The pattern of AQP5 expression likely explains this finding, as modest AQP5 staining is seen throughout BRAF$^{V600E}$-driven tumors as previously reported (*Trejo et al., 2013*), whereas BRAF$^{V600E}$/PI3Kα$^{H1047R}$-driven tumors show strong AQP5 in some areas and essentially absent AQP5 in other areas (*Figure 5A*). Co-immunostaining of AQP5 and LYZ showed similar patterns in which BRAF$^{V600E}$-driven tumors display widespread expression of both of AT1 and AT2 markers (*Figure 5C*). By contrast, BRAF$^{V600E}$/PI3Kα$^{H1047R}$-driven tumors show some areas that are double positive for both AQP5 and LYZ (*Figure 5C*, cyan arrowheads), some areas with only expression of AQP5 (*Figure 5C*, green arrowheads), some areas with only expression of LYZ (*Figure 5C*, red arrowheads), and some areas where neither is expressed (*Figure 5C*, yellow arrowheads). Quantification of these data shows correlation (Spearman Rho = 0.54) between AQP5 and LYZ in BRAF$^{V600E}$-driven tumors (*Figure 5D*), but lower correlation (Spearman Rho = 0.13) between these markers in BRAF$^{V600E}$/PI3Kα$^{H1047R}$-driven tumors (*Figure 5E*). Comparing these two tumor types directly shows a significant effect of genotype on staining distribution (*Figure 5F*), with BRAF$^{V600E}$/PI3Kα$^{H1047R}$-driven tumors showing a strong decrease in AQP5$^+$/LYZ$^+$ double positive cells, with corresponding increases in each of the remaining classes of cells (AQP5$^+$/LYZ$^-$; AQP5$^-$/LYZ$^+$; and AQP5$^-$/LYZ$^-$).

Co-expression of AT1 or AT2 markers in BRAF$^{V600E}$-driven tumors is reminiscent of what is observed in bipotent progenitor cells, which are reported to give rise to both AT1 and AT2 cells (*Desai et al., 2014*). To search for additional indicators of a bipotent progenitor like state induced by BRAF$^{V600E}$→MEK→ERK signaling, we analyzed transmission electron micrographs of BRAF$^{V600E}$-driven lung tumor sections from suitably manipulated *Braf$^{CA}$* mice 11 weeks p.i. Normal AT2 cells display a cuboidal morphology and contain many surfactant rich lamellar bodies (*Figure 5—figure supplement 1A*, cyan). BRAF$^{V600E}$-driven tumor cells displayed gross morphological similarities to AT2 cells (*Figure 5—figure supplement 1B*), but in a subset of tumor cells large vacuoles, not seen in normal AT2 cells, were observed (*Figure 5—figure supplement 1B*, purple). Enhanced magnification of these structures demonstrated a rough chrysanthemum like pattern of electron dense material (*Figure 5—figure supplement 1C*) characteristic of glycogen storage (*Revel, 1960*). As glycogen storage vacuoles are another hallmark of bipotent progenitor cells (*Desai et al., 2014*), we propose that BRAF$^{V600E}$→MEK→ERK signaling alone drives AT2 pneumocytes toward this fate. As BRAF$^{V600E}$/PI3Kα$^{H1047R}$-driven tumors show some cells with co-expression of AT1 and AT2 markers, and many cells with reductions of either or both of these marker classes, we believe the coincident

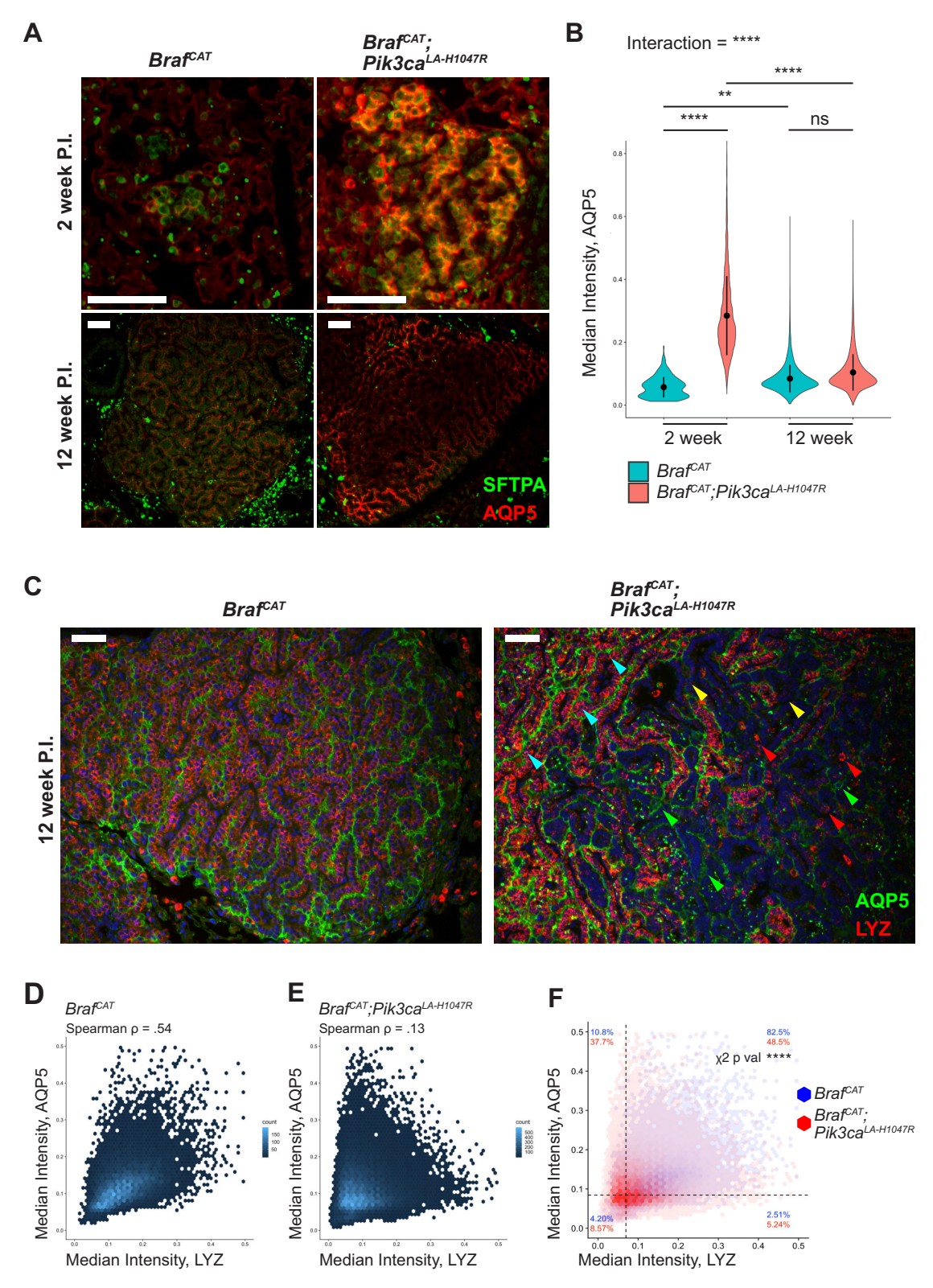

**Figure 5.** BRAF[V600E]/PI3Kα[H1047R] and BRAF[V600E] driven tumors both show effects on differentiation status, with BRAF[V600E]/PI3Kα[H1047R] driven tumors displaying more profound effects on identity. (**A**) BRAF[V600E]/PI3Kα[H1047R] and BRAF[V600E] driven hyperplasia both show immunoreactivity of the AT1 marker, AQP5, 2 weeks post initiation with BRAF[V600E]/PI3Kα[H1047R] driven hyperplasia showing enhanced immunostaining. BRAF[V600E]/PI3Kα[H1047R] and BRAF[V600E] driven tumors also show immunoreactivity of AQP5 12 weeks post initiation with BRAF[V600E]/PI3Kα[H1047R] driven tumors showing a more

*Figure 5 continued on next page*

*Figure 5 continued*

variable pattern of immunostaining. Scale bars = 100 um. (B) Quantitation demonstrating significant effect of PI3Kα$^{H1047R}$ on AQP5 immunoreactivity in BRAF$^{V600E}$ driven tumors 2 weeks post initiation. No difference seen in AQP5 immunoreactivity between BRAF$^{V600E}$/PI3Kα$^{H1047R}$ and BRAF$^{V600E}$ driven tumors 12 weeks post initiation. This appears to be the result of a slight increase in AQP5 immunoreactivity between 2 and 12 weeks in BRAF$^{V600E}$ driven tumors and a more dramatic decrease in AQP5 immunoreactivity between 2 and 12 weeks in BRAF$^{V600E}$/PI3Kα$^{H1047R}$ driven tumors. ANOVA p<1e-5, multiple comparisons done by Tukey's Honest Significant Difference test, ****: p<1 e −5, **p=0.0014. (C) BRAF$^{V600E}$ driven tumors display widespread immunoreactivity to both AQP5 and the AT2 marker, LYZ, 12 weeks post initiation. BRAF$^{V600E}$/PI3Kα$^{H1047R}$ driven tumors show cells with widely varied expression of differentiation markers, including AQP5+, LYZ+ (Cyan arrows); AQP5-, LYZ+ (Red arrows); AQP5+, LYZ- (Green arrows); and AQP5-, LYZ- (Yellow arrows) cells. Scale bars = 100 um. (D) BRAF$^{V600E}$ driven tumors show relatively high association between AQP5 and LYZ immunoreactivity (Rho = 0.54). (E) BRAF$^{V600E}$/PI3Kα$^{H1047R}$ driven tumors show relatively low association between AQP5 and LYZ immunoreactivity (Rho = 0.13) (F) Overlay of BRAF$^{V600E}$/PI3Kα$^{H1047R}$ and BRAF$^{V600E}$ driven tumors 12 weeks post initiation. Dashed line for each marker drawn at mean - one standard deviation of BRAF$^{V600E}$ driven tumors. BRAF$^{V600E}$/PI3Kα$^{H1047R}$ driven tumors show fewer AQP5+, LYZ + cells, most strongly accounted for by an increase in AQP5+, LYZ- cells, but with increases also seen in AQP5-, LYZ + and AQP5-, LYZ- cells. Chi square test associates genotype with distribution, p val <1e-5.

DOI: https://doi.org/10.7554/eLife.43668.025

The following source data, source code and figure supplements are available for figure 5:

**Source code 1.** R script to perform statistics on *Figure 4—source data 1*, as well as plot these results.
DOI: https://doi.org/10.7554/eLife.43668.027
**Source code 2.** Cellprofiler pipeline to quantify raw images from BRAF$^{V600E}$/PI3Kα$^{H1047R}$ and BRAF$^{V600E}$ driven tumors, producing *Figure 5—source data 1*.
DOI: https://doi.org/10.7554/eLife.43668.028
**Source data 1.** Cellprofiler output quantifying AQP5 and LYZ immunofluorescence in BRAF$^{V600E}$/PI3Kα$^{H1047R}$ and BRAF$^{V600E}$ driven tumors.
DOI: https://doi.org/10.7554/eLife.43668.029
**Figure supplement 1.** BRAF$^{V600E}$-driven lung tumors show characteristics of bipotent progenitor cells.
DOI: https://doi.org/10.7554/eLife.43668.026

activation of ERK1/2 plus PI3'-lipid signaling promotes more profound de-differentiation of AT2 cells.

## PGC1α expression correlates with AT2 marker expression

To identify novel candidate transcription factors that may participate in the establishment or maintenance of AT2 pneumocyte identity and function (*Figure 6A*), we took a three-step approach. First, we performed whole transcriptome correlation network analysis using Weighted Gene Correlation Network Analysis (WGCNA) (*Langfelder and Horvath, 2008*) to identify potentially relevant gene modules (*Figure 6B*). The majority of the AT2-100 and AT1-100 mRNAs fell into a single cluster (Cluster 2 – Dark blue *Figure 6B,C*). Next, to identify candidate regulators within cluster 2, we filtered the 2852 mRNAs in this cluster to select for those with demonstrated roles in transcriptional regulation. Finally, we filtered these selected transcription factors for differential expression in BRAF$^{V600E}$- vs BRAF$^{V600E}$/PI3Kα$^{H1047R}$-driven lung tumors using a promiscuous cutoff of adjP <0.2. The intersection of these three methods left a single candidate, the nuclear receptor co-activator, PGC1α (*Figure 6D*). PGC1α is a known transcriptional regulator, clusters with the majority of AT1 and AT2 genes, and its mRNA is significantly decreased in BRAF$^{V600E}$/PI3Kα$^{H1047R}$-driven tumors compared to BRAF$^{V600E}$-driven tumors (*Figure 6E*). The control of *PPARGC1A* levels by PI3'-lipid signaling may also be true in human lung tumors: those bearing mutations in either *PIK3CA*, *AKT1*, or *PTEN* have significantly reduced *PPARGC1A* mRNA expression compared to tumors bearing none of these mutations (*Figure 6—figure supplement 1A*). Finally, we sought to determine if PGC1α expression correlates with maintenance of lung identity on a cell-by-cell basis within BRAF$^{V600E}$/PI3Kα$^{H1047R}$-driven tumors in mice. Immunostaining revealed that cells with decreased expression of SFTPA lack nuclear PGC1α (*Figure 6F*, red arrows). Conversely, tumor cells with detectable nuclear localization of PGC1α display readily detectable levels of SFTPA (*Figure 6F*, green arrows).

## Silencing of PGC1α expression promotes de-differentiation of BRAF$^{V600E}$-driven lung tumors

To directly test if PGC1α regulates AT2 identity, we crossed a floxed, conditional null allele of *Pgc1α* (*Ppargc1a$^{lox/lox}$*) to *Braf$^{CAT}$* mice and generated littermate cohorts of *Braf$^{CAT}$; Ppargc1a$^{lox/lox}$*, and

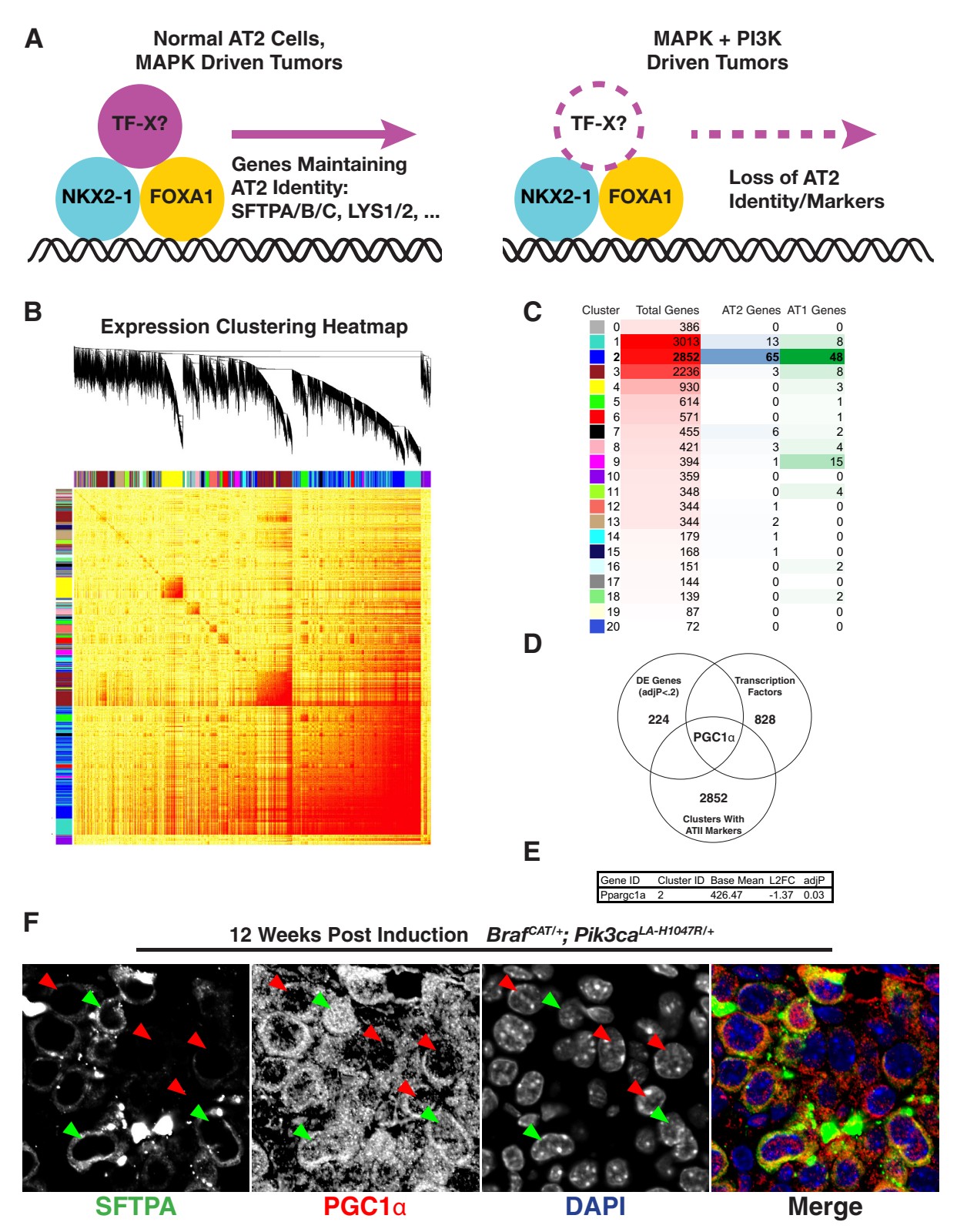

**Figure 6.** Loss of PGC1α Expression Correlates With Change in Expression of Lung Markers. (**A**) Hypothetical model showing an additional factor cooperating with lung lineage transcription factors that is downregulated upon dual arm mutational activation of growth factor signaling. (**B**) Weighted gene correlation network analysis (WGCNA) heat map identifies 21 correlated gene expression modules. Gene tree shows relationship of individual genes, where multi-color bars adjacent to heat map identify individual clusters. (**C**) Table summarizing result of WGCNA analysis and AT1/AT2

*Figure 6 continued on next page*

*Figure 6 continued*

memberships. Cluster number is listed adjacent to color corresponding to cluster in (**B**). For each cluster, shown is the total number of genes along with the number of AT1 and AT2 marker genes from AT1-100 and AT2-100. Cluster two contains the majority of both AT1 and AT2 marker genes. (**D**) A three factor approach to identify novel regulators of pneumocyte identity. Within the intersection of differentially expressed genes, genes co-regulated with the majority of AT1 and AT2 specific genes, and known transcription factors, lies a single gene, PGC1α. (**E**) PGC1α is significantly downregulated in BRAF$^{V600E}$/PI3Kα $^{H1047R}$ driven tumors compared to BRAF$^{V600E}$ driven tumors; adjP is Benjamini-Hochberg corrected P value from DESeq2. (**F**) Decreased nuclear PGC1α immunoreactivity (red arrows) correlates with loss of AT2 identity on a cell by cell basis. AT2 identity is maintained in those cells which maintain nuclear PGC1α immunoreactivity (green arrows).

DOI: https://doi.org/10.7554/eLife.43668.030

The following source data, source code and figure supplements are available for figure 6:

**Source code 1.** R script to perform weighted correlation network (WGCNA) on *Figure 6—source data 1*.

DOI: https://doi.org/10.7554/eLife.43668.033

**Source data 1.** DEseq2 normalized RNA-seq count output of all BRAF$^{V600E}$/PI3Kα$^{H1047R}$ and BRAF$^{V600E}$ driven tumors.

DOI: https://doi.org/10.7554/eLife.43668.034

**Figure supplement 1.** Mutations predicted to affect the PI3' lipid signaling pathway correlate with decreased *PPARGC1A* in human lung adenocarcinoma.

DOI: https://doi.org/10.7554/eLife.43668.031

**Figure supplement 1—source data 1.** *Ppargc1a* FPKM values from human tumors.

DOI: https://doi.org/10.7554/eLife.43668.032

*Braf$^{CAT}$; Ppargc1a$^{lox/+}$* mice. Lung tumorigenesis was initiated in these mice using 10$^6$ pfu Ad5-*Sftpc-CRE* (*Figure 7A*) with lungs harvested 12 weeks p.i. for isolation and analysis of tdTomato$^+$ tumor cells by RNA sequencing. As expected, BRAF$^{V600E}$/PGC1α$^{Null}$-driven lung tumor cells showed decreasd expression of mRNAs encoding proteins involved in oxidative phosphorylation (*Figure 7A*). BRAF$^{V600E}$/PGC1α$^{Null}$-driven tumors also displayed a widespread decrease in markers of AT2 differentiation status as compared to BRAF$^{V600E}$-driven tumors that retain PGC1α expression (*Figure 7A* and *Figure 7—figure supplement 1*). Interestingly, silencing of PGC1α recapitulated some other aspects of PI3Kα activation, including increasing markers of EMT, but some noticeable differences were observed in mRNA profiles including significant increases in ciliated and club cell markers (*Figure 7A*). LYZ and SFTPA immunoreactivity in BRAF$^{V600E}$ and BRAF$^{V600E}$/PGC1α$^{Null}$-driven tumors validated the decrease in AT2 marker expression in the absence of PGC1α expression (*Figure 7B–C*, data from *Figure 7—source data 1*).

## PGC1α cooperates with NKX2-1 and FOXA1 to transactivate AT2 promoters

We next sought to test if the role of PGC1α in AT2 pneumocyte identity maintenance could be through direct action on the promoters of AT2 pneumocyte specific genes. As PGC1α generally co-activates nuclear receptors such as PPARγ, we sought to discover its potential binding partner relevant to AT2 pneumocyte gene regulation. To this end we performed a motif discovery analysis using Multiple Em for Motif Elicitation (MEME) (*Bailey and Elkan, 1994*; *Bailey et al., 2009*) algorithm and scanning the promoter regions of the AT2-100. The most enriched novel motif (*Figure 7—figure supplement 1B–C*) bears significant similarity to the known binding motif of the nuclear receptor, NR5A2 (JASPAR: MA0505.1), also known as Liver Receptor Homolog (LRH) 1 (*Gupta et al., 2007a*).

To functionally test whether PGC1α can act at promoter sequences to regulate key markers of AT2 identity, we generated reporter constructs in which ~ 5 kb upstream of the transcription start site (TSS) of the genes encoding surfactant proteins A, B, or C (*Sftpa/b/c*) was inserted into a luciferase reporter plasmid. Transfection of individual expression constructs for NR5A2, PGC1α, NKX2-1, or FOXA1 showed modest promoter induction of up to four fold over mock-transfected cells (*Figure 7D*). However, co-transfection of these four factors together showed a dramatic 25–100 fold induction of the SFTPA, SFTPB, and SFTPC promoters. Importantly, the absence of PGC1α severely hampered the ability of NKX2-1 and FOXA1 to transactivate the surfactant A and B promoters, suggesting a functional role of PGC1α in the transcriptional transactivation of AT2 promoters.

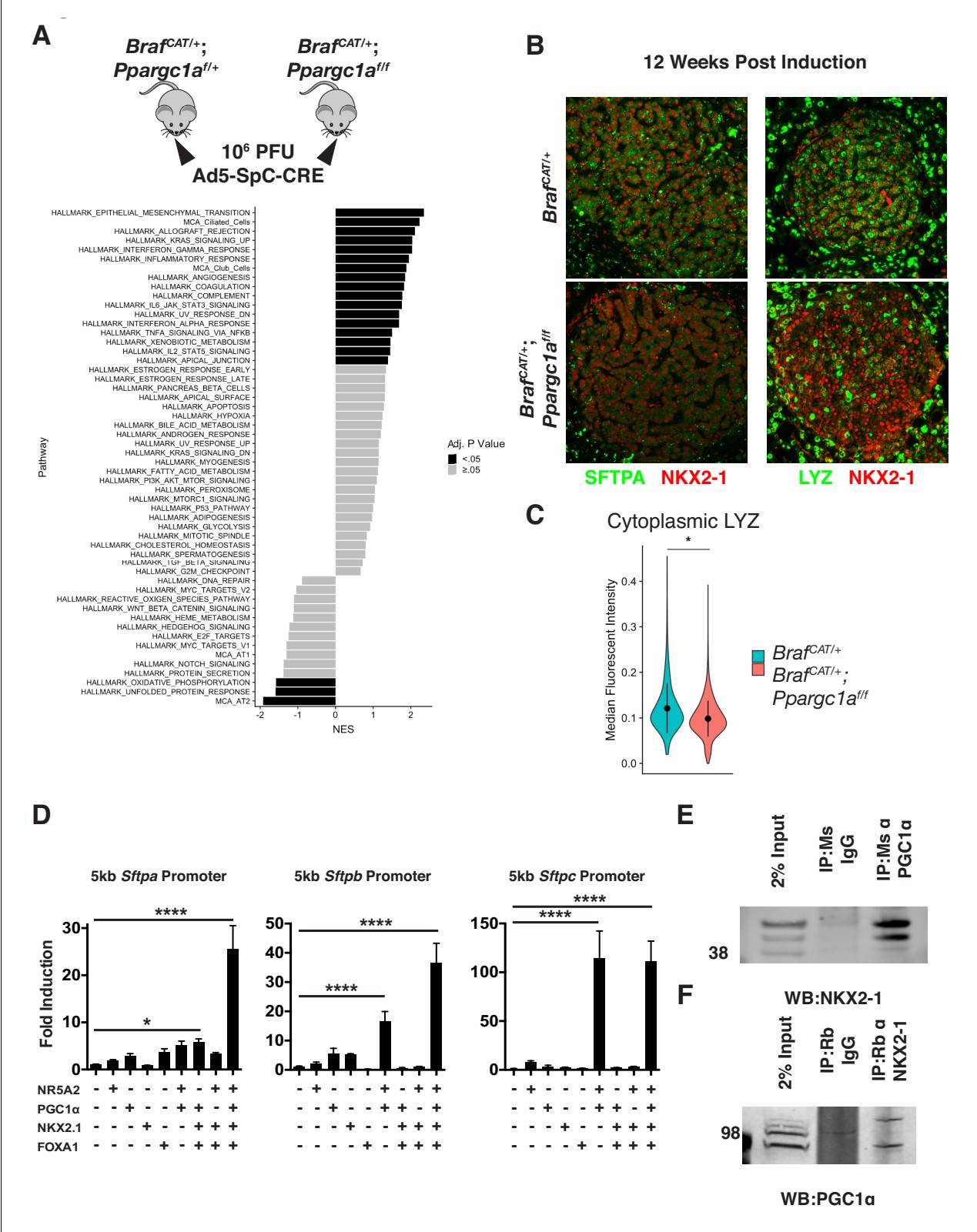

**Figure 7.** PGC1α is required for maintenance of lung identity in BRAF[V600E] driven tumors. (**A**) Tumors were induced in cohorts of *Braf*[CAT/+] and *Braf*[CAT/+];*Ppargc1a*[f/f] mice via intranasal instillation of 10[6] PFU Ad5-SpC-CRE and harvested from each genotype 12 weeks post tumor induction via tissue dissociation and FACS. GSEA analyses of hallmark pathways and lung identity gene sets. Black bars indicate adjP <.05, gray bars indicate Benjamini-Hochberg corrected enrichment statistic adjP ≥. 05. (**B**) Immunostaining confirms decreased expression of the AT2 markers SFTPA and LYZ

*Figure 7 continued*

in BRAF<sup>V600E</sup>/PGC1α<sup>NULL</sup> tumors. (**C**) Quantitation demonstrating a significant decrease of LYZ immunoreactivity in BRAF<sup>V600E</sup>/PGC1α<sup>NULL</sup> tumors. Wilcoxon rank sum p val. = 0.0288. (**D**) Luciferase assays in HEK293T cells demonstrating the cooperation of NKX2-1, FOXA1, PGC1α, and NR5A2 in transactivation of surfactant promoters. All three promoters showed significant induction by ordinary one-way ANOVA (p<0.0001). Comparison of individual groups to mock transfected controls by Dunnett's test for multiple comparisons: (*) p=0.0189, (****) p<0.0001. (**E**) Co-Immunoprecipitation of NKX2-1 by immunoprecipitation with a mouse monoclonal antibody recognizing PGC1α but not with IgG. (**F**) Co-Immunoprecipitation of PGC1α by immunoprecipitation with a mouse monoclonal antibody recognizing NKX2-1 but not with mouse IgG.

DOI: https://doi.org/10.7554/eLife.43668.035

The following source data, source code and figure supplements are available for figure 7:

**Source code 1.** R script to perform gene set enrichment analysis on *Figure 7—source data 1*, as well as plot these results.
DOI: https://doi.org/10.7554/eLife.43668.038

**Source code 2.** R script to perform statistics on *Figure 7—source data 2*, as well as plot these results *eLife's* transparent reporting form.
DOI: https://doi.org/10.7554/eLife.43668.039

**Source data 1.** DEseq2 output of differentially expressed genes comparing BRAF<sup>V600E</sup>/PGC1α<sup>NULL</sup> and BRAF<sup>V600E</sup>/PGC1α<sup>HET</sup> driven tumors.
DOI: https://doi.org/10.7554/eLife.43668.040

**Source data 2.** Cellprofiler output quantifying immunofluorescence of LYZ in BRAF<sup>V600E</sup>/PGC1α<sup>NULL</sup> and BRAF<sup>V600E</sup>/PGC1α<sup>WT</sup> driven tumors.
DOI: https://doi.org/10.7554/eLife.43668.041

**Source data 3.** Data from luciferase assays looking for transactivation of *Sftpa*, *Sftpb*, and *Sftpc* promoters.
DOI: https://doi.org/10.7554/eLife.43668.042

**Figure supplement 1.** PGC1α is required for maintenance of lung identity in BRAF<sup>V600E</sup> driven tumors.
DOI: https://doi.org/10.7554/eLife.43668.036

**Figure supplement 2.** Proposed mechanism of PI3'-kinase-α promoting de-differentiation of lung tumors initiated by the BRAF<sup>V600E</sup> oncoprotein kinase.
DOI: https://doi.org/10.7554/eLife.43668.037

## PGC1α is found in complex with NKX2-1 in surfactant expressing cells

Based on the discovery that NR5A2 and PGC1α can cooperate with NKX2-1 and FOXA1 to transactivate AT2 pneumocyte promoters, we hypothesized that these proteins may exist in a biochemical complex. To test this directly, we used the immortalized mouse lung epithelial line MLE-12 (*Wikenheiser et al., 1993*). We first verified that these cells indeed express the pertinent proteins and maintain surfactant expression (*Figure 7—figure supplement 1D*). We next performed co-immunoprecipitation assays from MLE-12 cell extracts. While magnetic beads conjugated to mouse IgG did not enrich eluates for NKX2-1, magnetic beads conjugated to a mouse monoclonal antibody directed against PGC1α readily co-immunoprecipitated NKX2-1 (*Figure 7E*). To confirm this result, we showed that, while magnetic beads conjugated to rabbit IgG did not enrich eluates for PGC1α, magnetic beads conjugated to a rabbit monoclonal antibody directed against NKX2-1 also co-immunoprecipitated PGC1α (*Figure 7F*). Thus, PGC1α and NKX2-1 appear to co-exist in a complex in cells that express genes encoding AT2 expressed surfactant proteins.

## Discussion

Genetically engineered mouse models have proven to be invaluable tools that complement the significant advances being made in the genetic and biochemical characterization of the initiation, progression and maintenance of human cancers. As genome sequencing efforts catalog clinically actionable mutations and their correlations with the cancer's observed phenotypes, directly testing how these mutations affect disease initiation, progression, and response to novel therapeutics becomes a high priority. Here we describe a new GEM model that has allowed a deeper dissection of the stages of BRAF<sup>V600E</sup>-driven lung cancer, a disease that kills ~4000 patients per year in the U.S. A. (*Siegel et al., 2016*). To engineer this mouse we employed a strategy in which the previously targeted CRE-activated *Braf<sup>CA</sup>* allele was re-targeted to allow for the expression of both BRAF<sup>V600E</sup> and tdTomato from a single bicistronic mRNA following CRE-mediated recombination. Consequently, all BRAF<sup>V600E</sup> expressing cells are predicted to be red fluorescent by virtue of co-expression of tdTomato. This is an alternative approach to the more widely employed method of using a fluorophore expressed in trans from a different promoter (*e.g. Rosa26*) the expression of which is co-activated by CRE recombinase (*Livet et al., 2007*; *Madisen et al., 2010*; *Muzumdar et al., 2007*;

*Snippert et al., 2010*). The approach described here has both advantages and disadvantages compared to the use of a fluorophore expressed in trans approach. The main disadvantage of our approach appears to be the modest fluorescence emission of tdTomato when expressed downstream of a P2A element from the mutationally activated $Braf^{T1910A}$ mRNA. Many LSL-XFP alleles have been designed for very high levels of expression by the incorporation of enhancers and strong promoters, whereas our model is driven by the endogenous *Braf* enhancer/promoter sequences. Hence, although the level of tdTomato fluorescence in lung tumors arising in $Braf^{CAT}$ mice is readily detected by flow cytometry, detection of tdTomato expression by immunohistochemistry in FFPE tissue sections is problematic. However, we have developed protocols that allow for detection of native tdTomato fluorescence in frozen sections (*Figure 1D*) or by a ligation proximity assay in FFPE sections (Daphne Pringle, unpublished). By contrast, the use of a fluorophore expressed in trans approach inherently relies on the simultaneous activity of CRE at distinct regions of the genome. Thus it is possible for the proto-oncogenic $Braf^{CA}$ locus to be subject to CRE-mediated recombination without recombination of the reporter, or vice-versa. While this is acceptable for some applications, it adds confounding noise that may be amplified under conditions when tumor cells have been specifically depleted by the application of pathway targeted or immunomodulatory therapy that has negligible effect on normal cells. Our approach avoids this noise entirely and results in a level of fluorescence that is directly correlated with expression of BRAF$^{V600E}$ on a cell-by-cell basis (*van Veen et al., 2016*). Importantly, our approach is especially compatible with the emerging technology of single cell RNA sequencing (scRNA-Seq), in which it is desirable to isolate single BRAF$^{V600E}$ oncoprotein kinase expressing cells for analysis.

Our studies were aided greatly by recent characterizations of the gene expression changes accompanying development of the mouse distal lung epithelium (*Treutlein et al., 2014*) and KRAS$^{G12D}$-mediated oncogenic transformation of AT2 cells (*Desai et al., 2014*). Using scRNA-Seq, the authors reconstructed the lineage hierarchies of AT1, AT2, club and ciliated cells, as well as provided a new list of markers useful in identifying these major cell types of the distal lung epithelium. We used these marker lists extensively and without bias or selection for most of our analyses. Importantly, we constructed gene sets for the GSEA analyses that showed widespread decreased AT2 marker expression in BRAF$^{V600E}$/PIK3CA$^{H1047R}$-driven lung tumors. Analysis of the full complement of AT2 markers paints a dramatic picture; nearly every AT2 identity marker is diminished in expression in BRAF$^{V600E}$/PIK3CA$^{H1047R}$-driven tumors as compared to BRAF$^{V600E}$-driven tumors, as early as two weeks p.i. Importantly we noted a similar early and consistent repression of PGC1α expression in BRAF$^{V600E}$/PIK3CA$^{H1047R}$-driven tumors. Here our data support previous findings that indicate that PGC1α expression is repressed by the PI3K→AKT signaling axis via three insulin response sequences found in the PGC1α promoter shown to bind to the AKT-regulated FOXO1 transcription factor (*Daitoku et al., 2003*; *Kemper et al., 2014*). It has also been suggested that BRAF$^{V600E}$ signaling can suppress PGC1α expression via the MITF transcription factor in melanoma, further demonstrating the complicated interplay between growth factor signaling and oxidative phosphorylation (*Haq et al., 2013*), though this study did not examine any potential connection of PGC1α to tumor cell differentiation status.

*PIK3CA* mutation is found in ~4% of human LUAD (*Campbell et al., 2016*) that, although quite rare, still represents a significant patient population due to the high prevalence of lung cancer in our society. Previous studies have demonstrated that the ability of mutationally-activated KRAS$^{G12D}$ to bind and activate PI3Kα is critical for its tumor promoting activities, as well as tumor maintenance (*Castellano et al., 2013*; *Gupta et al., 2007b*; *Molina-Arcas et al., 2013*; *Murillo et al., 2018*). Despite this, activation of PI3'-lipid signaling remains rate limiting with respect to tumor initiation and growth in KRAS$^{G12D}$-driven lung tumors as evidenced by the strong cooperation between KRAS$^{G12D}$ and PIK3CA$^{H1047R}$ in promoting lung tumorigenesis in a GEM model (*Green et al., 2015*). This cooperation likely reflects the fact that the initial expression of KRAS$^{G12D}$ is a relatively weak activator of both PI3'-lipid signaling and the RAF→MEK→ERK pathways (*Cicchini et al., 2017*). Indeed, whereas KRAS$^{G12D}$ derived lung cancer cells show little phosphorylation of AKT at a key activating amino acid (S473), KRAS$^{G12D}$/PIK3CA$^{H1047R}$ derived cells show strong pS473-AKT phosphorylation (*Green et al., 2015*). Our results suggest that the lack of PI3'-lipid signaling in KRAS$^{G12D}$-driven lung tumors is not only limiting in tumor growth, but in propensity for AT2 pneumocyte de-differentiation. Intriguingly, recent studies have shown that stromally derived IGF-1 promotes a cancer stem cell like phenotype in *Kras* mutated cell lines (*Chen et al., 2014*), via the

PI3K→AKT signaling axis. It will be interesting to examine in future studies how various tumor genotypes and tumor microenvironments converge upon the initiation and evolution of LUAD de-differentiation.

Lung adenocarcinoma differentiation status, as judged by pathological criteria, remains a critically important prognostic factor in predicting patient survival (*Yoshizawa et al., 2011*). However, only in recent years have we begun to understand the genetic aberrations that can directly promote loss of differentiation status. The protein most directly demonstrated to influence both lung adenocarcinoma differentiation status and progression is NKX2-1, a homeodomain transcription factor. Indeed, in the *Kras*^LSL-G12D GEM model of KRAS^G12D-driven lung adenocarcinoma, NKX2-1 expression is diminished or silenced in the most poorly differentiated tumors. Moreover, shRNA-mediated inhibition of NKX2-1 expression is reported to enhance the metastatic potential of KRAS^G12D/TP53^Null-driven lung cancer cells (*Winslow et al., 2011*). Interestingly, concomitant expression of KRAS^G12D with genetic silencing of NKX2-1 expression has qualitatively different effects, promoting trans-differentiation of tumor cells into a gastric fate resembling de-differentiated mucinous adenocarcinoma (*Snyder et al., 2013*), an effect that requires the activity of FOXA1/FOXA2 (*Camolotto et al., 2018*). Intriguingly, while even haploinsufficiency of NKX2-1 promoted the appearance of mucinous adenocarcinoma in the *Kras*^LSL-G12D GEM model of lung adenocarcinoma, the same was not true in a lung adenocarcinoma model driven by expression of a mutationally-activated form of the EGF receptor (*Maeda et al., 2012*). In this model, haploinsufficiency of NKX2-1 slowed tumor progression rather than enhancing it. Importantly, loss-of-function mutations or silencing of NKX2-1 is a relatively infrequent event in lung adenocarcinoma. Instead NKX2-1 is often found as the most significantly focally amplified locus (*Campbell et al., 2016*). Further, in human NSCLC cell lines in which *NKX2-1* is amplified, RNAi-mediated inhibition of NKX2-1 expression elicited decreased cell division and apoptosis (*Kwei et al., 2008*). Hence, the role of NKX2-1 in LUAD progression is thus simultaneously critically important and also complicated. In this case, GEM models provide an ideal system in which to study the contribution of individual mutations to each step of tumor initiation and progression without the complications of mutagen-induced genome hypermutation as is common in *KRAS*-mutated human lung cancer cells (*Chalmers et al., 2017*). While a wealth of literature has shown the direct effect of NKX2-1 on AT2 promoters (*Bruno et al., 1995*), and the importance of NKX2-1 in normal lung development (*DeFelice et al., 2003*), it has also been shown that transcriptional co-activators are critical for NKX2-1 function (*Cassel et al., 2002*; *Di Palma et al., 2003*; *Park et al., 2004*; *Yi et al., 2002*). Here we have shown that the binding motif of the nuclear receptor, NR5A2 is highly enriched in the promoters of AT2 specific genes. We also demonstrate that NR5A2, and its known co-factor PGC1α (*Yazawa et al., 2010*), can potently enhance the activity of NKX2-1 at the promoters for surfactant proteins A and B. Interestingly, while NR5A2 and PGC1α can activate the promoter of SFTPC alone, the added presence of NKX2-1 and FOXA1 does not further co-activate this promoter. It may be the case that PGC1α and NKX2-1 act independently at this promoter, or it may be that there are additional or alternative transcriptional regulators not present in 293 T cells which, when present, allow PGC1α and NKX2-1 to cooperate. In vivo, in BRAF^V600E/PIK3CA^H1047R-driven tumors, we observed an early decrease of PGC1α mRNA expression, and importantly, a correlation between decreased PGC1α and SFTPA on a cell-by-cell basis within tumors. Combined with functional data in GEM models in which PGC1α expression was genetically silenced, these data argue that mutational-activation of PI3'-lipid signaling in BRAF^V600E-driven LUAD leads to diminished PGC1α expression, and that this reduced expression compromises the ability of NKX2-1/FOXA1 to maintain AT2 pneumocyte identity (*Figure 7—figure supplement 2A*). It is important to note that while genetic silencing of PGC1α recapitulated some aspects of PI3K activation in BRAF^V600E-driven driven lung tumors, there were interesting differences, including an increase in markers of club and ciliated cell identity, and no significant effect on AT1 marker expression. It is therefore highly likely that amongst the pleiotropic effects of PI3'-lipid signaling, inhibition of PGC1α-mediated signaling represents one of many important effector pathways. This novel mechanism of lung identity regulation is made more important by the observation that it can be induced by mutational-activation of PI3Kα in both KRAS^G12D- and BRAF^V600E-driven driven lung adenocarcinomas. Since the best characterized role of PGC1α is in the regulation of mitochondrial biogenesis, we were initially surprised to discover the cooperativity that PGC1α shows in the regulation of AT2 cell identity. However, it has recently emerged that crippling mitochondrial function via loss of the critical pyruvate transporter, MPC1, potently drives cells into a de-differentiated stem cell fate in drosophila and mouse intestinal

cells (*Schell et al., 2017*). These studies, and the data presented here, suggest exciting future studies to examine the role of PGC1α to act as a pleiotropic effector of tumor cell growth and differentiation state downstream of PI3'-lipid signaling.

# Materials and methods

## Key resources table

| Reagent type (species) or resource | Designation | Source or reference | Identifiers | Additional information |
|---|---|---|---|---|
| Gene (*Mus musculus*) | *Braf* | | Ensembl: ENSMUSG00000002413 | |
| Gene (*Mus musculus*) | *Pik3ca* | | Ensembl: ENSMUSG00000027665 | |
| Genetic reagent (*Mus musculus*) | *Braf*$^{CA}$ | McMahon lab stock | MGI:*Braf*$^{tm1Mmcm}$ JAX:017837 RRID:IMSR_JAX:017837 | |
| Genetic reagent (*Mus musculus*) | *Pik3ca*$^{H1047R}$ | Wayne Phillips | MGI:*Pik3ca*$^{tm1.1Waph}$ RRID:MGI:5427584 | |
| Genetic reagent (*Mus musculus*) | *Braf*$^{CAT}$ | This paper | N/A | New genetically engineered mouse reported in this paper |
| Genetic reagent (*Mus musculus*) | *Kras*$^{LSL}$ | The Jackson Lab | MGI:*Kras*$^{tm4Tyj}$ JAX:008179 RRID:IMSR_JAX:008179 | |
| Cell line (*Mus musculus*) | MLE-12 | ATCC | CRL-2110 RRID:CVCL_3751 | |
| Cell line (*Homo sapiens*) | 293T | Lab Stock | RRID:CVCL_0063 | |
| Antibody | Mouse monoclonal PGC1α | Millipore | Cat# 1F3.9 RRID:AB_10806332 | Co-IP, 10 ug WB, 1:500 |
| Antibody | Rabbit polyclonal PGC1α | Millipore | Cat# AB3242 RRID:AB_2268462 | IHC, 1:50 WB, 1:500 |
| Antibody | Rabbit monoclonal NKX2-1 | Abcam | Cat# AB76013 RRID:AB_1310784 | Co-IP, 10 ug IHC, 1:250 WB, 1:1000 |
| Antibody | Rabbit polyclonal Phospho-S327-NKX2-1 | CST | Cat# 13608 RRID:AB_2798273 | IHC, 1:250 |
| Antibody | Rabbit monoclonal FOXA1 | Abcam | Cat# AB23738 RRID:AB_2104842 | IHC, 1:250 |
| Antibody | Rabbit monoclonal FOXA1 | CST | Cat# 58613 RRID:AB_2799548 | WB, 1:2500 |
| Antibody | Rabbit monoclonal FOXA2 | CST | Cat# D56D6 RRID:AB_10891055 | IHC, 1:250 WB, 1:1000 |
| Antibody | Rabbit monoclonal SFTPA1 + 2 | Abcam | Cat# AB206299 RRID:AB_2810211 | IHC, 1:250 WB, 1:1000 |
| Antibody | Goat polyclonal SFTPA | Santa Cruz | Cat# SC-7699 RRID:AB_661292 | IHC, 1:100 WB, 1:1000 |
| Antibody | Goat polyclonal SFTPA | Santa Cruz | Cat# SC-7700 RRID:AB_661293 | IHC, 1:100 WB, 1:1000 |
| Antibody | Rabbit polyclonal SFTPA | Santa Cruz | Cat# SC-13977 RRID:AB_661294 | WB, 1:1000 |

*Continued on next page*

*Continued*

| Reagent type (species) or resource | Designation | Source or reference | Identifiers | Additional information |
|---|---|---|---|---|
| Antibody | Rabbit monoclonal Lysozyme | Abcam | Cat# AB108508 RRID:AB_10861277 | IHC, 1:250 WB, 1:1000 |
| Antibody | Goat polyclonal SFTPC | Santa Cruz | Cat# SC-7705 RRID:AB_2185505 | IHC, 1:250 WB, 1:1000 |
| Antibody | Rabbit polyclonal SFTPC | Santa Cruz | Cat# SC-13979 RRID:AB_2185502 | WB, 1:1000 |
| Antibody | Goat polyclonal CCA | Santa Cruz | Cat# SC-9772 RRID:AB_2238819 | IHC, 1:1000 |
| Antibody | Goat polyclonal AQP5 | Santa Cruz | Cat# SC-9890 RRID:AB_2059877 | IHC, 1:50 |
| Antibody | Rabbit polyclonal NR5A2 | Abcam | Cat# AB153944 RRID:AB_2810212 | WB, 1:1000 |
| Recombinant DNA reagent | M50 Super 8x TOPFlash | Addgene | Plasmid #12456 RRID:Addgene_12456 | Vector used to build SFTP Luciferase reporters |
| Recombinant DNA reagent | pCDNA3-mLRH1 | Holly Ingraham | | Ms NR5A2 expression vector |
| Recombinant DNA reagent | pDTA-TK | Addgene | Plasmid #22677 RRID:Addgene_22677 | Empty targeting vector for mouse production |
| Recombinant DNA reagent | MSCV-NKX2.1 | Addgene | Plasmid #31271 RRID:Addgene_31271 | NKX2.1 expression vector |
| Recombinant DNA reagent | PCDH-FOXA1 | Eric Snyder | | FOXA1 expression vector |
| Recombinant DNA reagent | FUW-mKate | This paper | | Fluorescent protein mKate expression vector used for assaying transfection efficiency |
| Recombinant DNA reagent | GFP-PGC1 | Addgene | Plasmid #4 RRID:Addgene_4 | PGC1α expression vector |
| Recombinant DNA reagent | pWZL-Hygro | Addgene | Plasmid #18750 RRID:Addgene_18750 | Empty vector used to normalize amount of DNA transfected |
| Recombinant DNA reagent | SFTPA-LUC | This paper | | 5 kb SFTPA promoter in luciferase reporter |
| Recombinant DNA reagent | SFTPB-LUC | This paper | | 5 kb SFTPB promoter in luciferase reporter |
| Recombinant DNA reagent | SFTPC-LUC | This paper | | 5 kb SFTPC promoter in luciferase reporter |
| Peptide, recombinant protein | TAT-CRE | Excellgen | Cat# Eg-1001 | Cell permeant CRE protein |
| Commercial assay or kit | Dynabeads protein G IP kit. | Thermo | Cat# 10007D | |
| Commercial assay or kit | Pierce firefly luc one step glow assay kit | Thermo | Cat# 16196 | |

*Continued on next page*

*Continued*

| Reagent type (species) or resource | Designation | Source or reference | Identifiers | Additional information |
|---|---|---|---|---|
| Software, algorithm | MEME | http://meme-suite.org | Multiple Em Motif Elucidation | |
| Software, algorithm | FIMO | http://meme-suite.org | Find Individual MOtifs | |
| Software, algorithm | GSEA | http://gsea.org | Gene Set Enrichment Analysis | |
| Software, algorithm | R | http://rstudio.com | R programming language | |
| Software, algorithm | Custom R scripts | github.com/jevanveen/vanveen-elife | Referenced scripts hosted at GitHub | |
| Other | Antigen Retrieval | *Syrbu and Cohen, 2011* | Antigen Retrieval Method for Immunostaining of Paraffin Sections | Method greatly aiding in immunostaining |

## Contact for reagent and resource sharing

Further information and requests for reagents should be directed to and will be fulfilled by the Lead Contact, Martin McMahon (martin.mcmahon@hci.utah.edu).

## Biological vs technical replication

Individual animals, tumors, and different cell lines comprise independent biological replicates. Repeated testing on the same cell line is considered technical replication.

## Experimental model: Mice

All mouse work was done with the approval of either the University of California IACUC under approval #AN089594 or the University of Utah IACUC under approval #15-11014. Mice were housed in microisolator cages on ventilated racks in AAALAC accredited vivaria. Mice were housed in groups, as possible, and were provided bedding enrichment. Animals were provided standard laboratory rodent chow or Capecchi's breeder diet when appropriate. Cages were supplied with water via a lixit system built into the housing rack or with a water bottle placed in the microisolator cage. Institute husbandry staff performed twice daily health checks. For breeding purposes, FVB/N and C57BL/6J mice were obtained from the Jackson Laboratory. In all animal experiments, animals were age matched to within 4 weeks of one another, all being between 3 to 4 months old. A power analysis was conducted in R to determine the number of animals to include based on RNA-Seq data of *Sftpa* from a previous experiment (power.t.test(delta = 2895.4, sd = 918, sig.level = 0.05, power = 0.8)) indicating 3.4 mice per group would be sufficient to see a 2-fold difference. This was rounded up and four mice were initiated for each time point and genotype. Equal numbers of male and female mice were selected for each time point and genotype. Mice were then randomized within sex and genotype and assigned to groups for harvest at different time points such that at each time point, 2 females and two males were euthanized for analysis. All mice used in these experiments were completely drug naïve and had never undergone other experimental procedures. Mice were on a mixed background of C57BL/6, 129, and FVB. Tumor initiation was performed by experimenters blinded to genotype.

## Experimental model: Cell lines

MLE-12 immortalized lung cells were newly obtained from ATCC and therefore assumed to be of the correct identity and to be free of mycoplasma contamination. MLE-12 cells were cultured in HITES medium supplemented with 2% FBS (HyClone) and penicillin/streptomycin. 293 T cells were from a frozen lab stock, whose identity was confirmed by STR profiling at the Huntsman Cancer Institute DNA sequencing core, and whose mycoplasma contamination status was confirmed to be

negative by PCR. 293 T cells were maintained in DMEM (Life Technologies) supplemented with 10% FBS (HyClone) and supplemented with 5 mM glutamine and penicillin/streptomycin.

## Vector construction

All PCR steps were performed with CloneAmp (Clontech) high-fidelity polymerase premix and all newly constructed vectors were verified by Sanger sequencing. The targeting vector used to produce the $BRaf^{CAT}$ mouse strain was built by the cloning of three fragments into the dual selection targeting vector pDTA-TK. Two fragments, comprising 4.8 kb and 3.8 kb targeting homology arms were amplified from a C57BL/6J BAC containing the mouse $BRaf$ locus. A third fragment comprising the entirety of the genetically engineered module was assembled by gene synthesis (Genewiz). The three fragments were cloned into the AGEI site of pDTA-TK using In-Fusion (Clontech).

New luciferase reporter constructs for the promoters of mouse surfactant proteins A, B, and C were created by amplification of mouse genomic DNA from a tail biopsy. Primer blast (NCBI) was used to create primers which captured 4500–5500 base pairs of promoter sequence beginning with the bases found immediately before the first annotated transcribed exon (ensembl.org). The TCF/LEF sites found in the M50 Super Top Flash construct were removed by digestion with KPNI and XHOI and In-Fusion was used to clone the SFTPA, B, or C promoters in their place. The FUW-Kate plasmid was constructed by gene synthesis (IDT) of sequence encoding the mKate2 red fluorophore with ends compatible for In-Fusion cloning into the EcoRI and BamHI sites of FUW.

## ES cell targeting and screening

The $BRaf^{CAT}$ targeting construct was linearized by restriction enzyme digestion with the rare cutting enzyme I-CeuI and electroporated into 2H1 $Braf^{CA/+}$ ES cells, which were selected for construct integration using puromycin. Three 96-well plates of resistant clones were screened via PCR using one primer specific to the targeting construct and one primer found in the mouse genome, outside of the construct homology arms, such that a 4.8 kb product would be the result of homologous targeting construct integration, whereas integration by NHEJ would yield no product. Cell permeant TAT-CRE was added directly to culture media at a final concentration of 1 uM to test for functionality and CRE dependence of the fluorophore. Purified TAT-CRE was obtained from Excellgen (Rockville, MD).

## Tumor induction

Mice were euthanized for analysis of lung tumor cell fluorescence at 2, 6, or 12 weeks p.i. Mice destined for lung harvest at either 2 or 6 weeks were initiated with $10^7$ pfu of Ad5-$Sftpc$-CRE (*Fasbender et al., 1998*; *Sutherland et al., 2014*). Mice destined for lung harvest at 12 weeks were initiated with a lower titer ($10^6$ pfu) of Ad5-$Sftpc$-CRE to avoid encountering premature endpoints due to tumor burden.

## Tissue harvest

At euthanasia, mice were perfused by first cutting the vena cava caudalis underneath the liver and then injection of DEPC treated PBS into the right ventricle of the heart until lungs turned white (*Arlt et al., 2012*). The cranial, medial and caudal lobes of the right lung were first taken and placed into ice cold PBS and placed on ice. A new syringe and needle containing 10% neutral buffered formalin (NBF) was inserted into the larynx of mice and 10 ml was slowly infused to initiate fixation of the remaining lung lobes. The two remaining partially fixed lung lobes were placed into 25 ml of NBF and incubated for 24 hr before being processed into paraffin and sectioned.

## Tissue dissociation and FACS

Tumor bearing lungs were minced using fine scissors in a 0.5 mg/ml solution of Liberase TM (Roche). Minced tissue was incubated in a 37°C degree water bath for 15 min before being dissociated by pipetting up and down with a 1 ml pipette tip. Red blood cell lysis was performed by the addition of BD PharmLyse to 1x concentration and samples were incubated for an additional 10 min. Samples were passed through a 100 um filter fitted on a 50 ml conical tube. Filters were rinsed with 9 ml of ice cold Hanks Balanced Salt Solution (HBSS) and flow through was pelleted by centrifugation. From this point on, cells were kept on ice until lysis. Pellets were resuspended in 10 ml ice cold HBSS and

passed through a 70 um filter affixed to the same 50 ml conical tube. Finally, dissociated cells were resuspended in 1 ml HBSS and passed through 35 um cell strainer cap into 5 mL round bottom polystyrene tubes. FACS was performed on a Becton-Dickinson ARIA III fitted with 100 uM nozzle, using gates as shown in the figures. tdTomato positive cells were sorted into RLT lysis buffer, homogenized and stored at −80˚C until RNA purification.

Library Construction and Sequencing cDNA libraries for RNA-Sequencing experiments were produced by the High Throughput Genomics Core at the Huntsman Cancer Center. RNA was purified using the Qiagen RNeasy micro system. RNA integrity was assayed using the Agilent TapeStation 2200 and High Sensitivity RNA ScreenTapes. RNA data was included in final analyses if RINe values were above 6. No other exclusion of samples was done. For all samples, libraries were prepared using the Nugen Ovation Ultralow Library System V2. Libraries were sequenced using an Illumina HiSeq 2000 device with 50 single end cycles and v4 chemistry.

## Bioinformatics

For RNA sequencing analysis, RNA transcript abundance was estimated using Salmon with default parameters on the main instance of the Galaxy webserver (https://usegalaxy.org). Differentially expressed genes were then determined by use of DESeq2 using default parameters on the main instance of the Galaxy webserver (https://usegalaxy.org), resulting in the datasets *Figure 2—source data 1*, *Figure 7—source data 1*. Gene set enrichment analysis was next performed using the R-package 'fgsea' with default parameters using the scripts provided as *Figure 2—source code 1*, *Figure 3—source code 1*, *Figure 7—source code 1*. Gene correlation network analysis was performed in the R-package 'WGCNA' with the following parameters: To decrease noise, genes were filtered for minimal expression (R-norm >40), leaving 14207 genes to be clustered. These genes were clustered in a single block with a soft-thresholding power of '3' as recommended in the WGCNA documentation based on the scale free topology fit index of our data. WGCNA R script provided as *Figure 6—source code 1*, data provided as *Figure 6—source data 1*. For motif discovery, the promoter regions from the top 100 most specific AT2 marker genes were defined as 5 kb upstream of the transcriptional start site, and downloaded from biomart (https://www.ensembl.org/biomart). This list of sequences was filtered by repeat masker to remove low complexity DNA (repeatmasker.org). The filtered list was analyzed for enriched novel motifs using MEME (http://meme-suite.org) with default settings less the following parameters: Find 25 motifs of width between 6 and 25 nucleotides. Novel motifs were matched to known transcription factor motifs from human and mouse (HOCOMOCO v11 full) using TomTom (http://meme-suite.org) with default settings. To determine if PI3'-lipid signaling strength affects *Ppargc1a* transcript levels in human tumors, lung adenocarcinoma data were downloaded from the NCI Genomic Data Commons Data Portal and segregated into those cases predicted to have strong or weak activation of PI3'-lipid signaling (*Figure 6—figure supplement 1—source data 1*). The groups were defined as such: 'Strong activation' due to mutation in at least one of the following genes: *Pik3ca*, *Pten*, *Pik3r1*, and *Akt1*, 'Unknown PI3K activation status' due to none of these mutations being detectable. FPKM values for *Ppargc1a* were then compared between these two groups. This resulted in n = 211 control samples and n = 19 mutant samples with predicted strong activation of PI3K signaling. All R scripts written for this study are available at GitHub (*van Veen, 2019*; copy archived at https://github.com/elifesciences-publications/vanveen-elife).

## Tissue processing and antibody staining

Harvested tissues were processed and embedded in paraffin, and sectioned at 4 uM. After deparaffinization in Citra-Clear (Stat Lab, McKinney, TX), sections were re-hydrated in an ethanol series and antigens were unmasked using heated incubation in Tris-EDTA-SDS (*Syrbu and Cohen, 2011*). Sections were blocked for non-specific interaction in 10% Normal Donkey Serum in PBS and antibody staining was performed using the following primary antibody concentrations: PGC1α (AB3242) 1:50. NKX2-1 (AB76013) 1:250. Phospho-S327-NKX2-1 (13608) 1:250. FOXA1 (AB23738) 1:250. FOXA2 (D56D6) 1:250. SFTPA (SC-7699) 1:250. Lysozyme (AB108508) 1:250. SFTPC (SC-7705) 1:250. CCA (SC-9772) 1:1000. AQP5 (SC-9890) 1:50. Primary antibody incubation was performed overnight at four degrees. After washing, alexa-488 and alexa-594 conjugated donkey anti mouse and donkey

anti rabbit secondary antibodies were diluted 1:250, and incubated on sections for 2 hr at room temperature. Stained sections were counterstained in DAPI and mounted in fluoromount G.

## Microscopy and quantitation of immunofluorescent images

For overview see *Figure 3—figure supplement 1*. Fluorescent imaging in *Figure 6d* was performed on a Zeiss Apotome. Fluorescent imaging in *Figure 5* was performed on a Leica DM1000. All other fluorescent imaging was performed on a Nikon Ti-E inverted microscope employing a high sensitivity Andor Clara CCD camera. All images being compared in figures and in quantification were captured at exactly the same parameters for light and exposure. Acquisition settings were set such that pixel intensities were below saturation within regions of interest. When image intensity scales were adjusted for clarity, the intensity scales of images compared were set at exactly the same input and output levels. Intensity scales were never modified before quantitation. Images of tumor bearing lungs were imported into NIH ImageJ, where individual tumors were traced, and matched TIFF files were exported for each available fluorescent channel. A custom pipeline was built in CellProfiler to identify tumor cells based first upon identifying tumor nuclei using NKX2-1 immunoreactivity, when available, or DAPI staining when NKX2-1 immunostaining had not been performed. Tumor cells were defined based on propagation from identified nuclei, and tumor cytoplasm was defined as tumor cell minus tumor nucleus. For tumors analyzed in *Kras* based models, tumors were more diffuse and intermingled with surrounding parenchyma and so identifying tumor cells based on propagation from nuclei led to poor performance, and so tumor cells were defined as a three pixel ring around the tumor nucleus. Measurements were taken from pertinent TIFF files within individual nuclear, cellular, and cytoplasmic objects. Data were exported into comma separated value files and imported into R Studio using a custom script. For the purposes of graphing, individual tumor cell points were graphed using a custom script employing the R function ggplot(). For all measurements, median fluorescence within each cellular object was the primary data output. For the purposes of quantification, tumor cells were not treated as independent, but whole tumor averages were considered (mean of median fluorescence values), and each tumor was treated as independent. Fluorescence was not assumed to fit a normal distribution, and as such two factor comparisons were done using Wilcoxon Rank Sum test to generate p values using the R function wilcoxon.test(). For comparisons of more than two conditions, one way ANOVA was performed using aov() followed by Tukey's honest significant difference using TukeyHSD(). Chi-Squared tests were performed in R with the function chisq.test(). When quadrants were drawn defining 'negative' and 'positive' staining: BRAF$^{V600E}$ driven tumors were noted to have relatively uniform positive expression of markers studied, and so 'negative' was defined as any tumor cell with median fluorescence less than one standard deviation below the mean of median fluorescence in BRAF$^{V600E}$ driven tumor cells. For all imaging and quantification, images were captured from 2 to 4 separate animals bearing tumors per group. Specific scripts and data for figures as follows: *Figure 3*: CellProfiler pipeline (*Figure 3—source code 3*) used with raw images to produce *Figure 3—source data 1*, *2* and *3*. *Figure 3—source data 1*, *2* and *3* then used with *Figure 3—source code 2* to perform statistics and produce graphs. *Figure 4*: CellProfiler pipeline (*Figure 4—source code 2* and *Figure 4—source code 3*) used with raw images to produce *Figure 4—source data 1* and *Figure 4—source data 2*. *Figure 4—source data 1* and *2* then used with *Figure 4—source code 1* to perform statistics and produce graphs. *Figure 5*: Cellprofiler pipeline (*Figure 5—source code 2*) used with raw images to produce *Figure 5—source data 1*. *Figure 5—source data 1* then used with *Figure 5—source code 1* to perform statistics and produce graphs. *Figure 7*: CellProfiler pipeline (*Figure 3—source code 3*) used with raw images to produce *Figure 7—source data 1*. *Figure 7—source data 1* then used with *Figure 7—source code 2* to perform statistics and produce graphs.

## Luciferase assays

Using Fugene 6, HEK293 cells were co-transfected with luciferase reporter constructs, candidate transcriptional regulators, a fluorescent reporter construct to measure transfection efficiency (FUW-Kate), and empty vector (pUC19) to standardize total amount of DNA transfected across conditions to 100 ng/well of 96 well plate. Transfection efficiency was measured by fluorescence on an Incucyte ZOOM automated microscopy system (https://www.essenbioscience.com/). Luciferase production was measured with the Pierce Firefly Luc One-Step Glow Assay Kit, normalized to transfection

efficiency, and represented as fold change over cells transfected with no candidate transcriptional regulators. All assays were performed in triplicate. Data provided as *Figure 7—source data 3*.

## Protein biochemistry

For immunoblot analysis of protein expression in MLE-12 cells, cells were lysed on ice in RIPA buffer supplemented with Thermo Halt protease inhibitor complex. Lysates were separated by acrylamide electrophoresis on pre-cast Novex 4–12% gradient Bis-Tris gels and transferred onto PVDF membranes using an Invitrogen iblot two transfer device. After blocking of non-specific interactions using Li-Cor blocking reagent, membranes were incubated in PBS containing antibodies at the following concentrations: SFTPC: (sc-13979) 1:1000; SFTPA: (sc-13977), 1:1000; NKX2-1: (ab76013), 1:1000; FOXA1: (58613), 1:2500; PGC1α: (ab3242), 1:500; NR5A2 (ab153944), 1:1000. Protocol was repeated twice for a total of n = 3 with equivalent results.

For co-immunoprecipitation experiments, 10 ug of the following antibodies were bound to Dynabeads magnetic Protein G beads (Ms anti-PGC1α (1F3.9), Rb anti-NKX2-1 (AB76013), Normal Rabbit IgG (CST#2729), Mouse IgG1 (CST#5415)) for 15 min at room temperature. All following steps were performed at four degrees centigrade. MLE-12 cells were lysed by 15 passages through a 25 gauge needle in the following buffer: 20 mM Tris HCl pH 8, 137 mM NaCl, 0.1%(v/v) Nonidet-P40. Lysates were centrifuged for 5 min at 1000xg to remove insoluble fraction. Cleared lysates were divided and incubated with either target antibody or IgG bound magnetic beads for 30 min before proceeding with washing and elution steps following product protocol. Eluates were separated by acrylamide electrophoresis on pre-cast Novex 4–12% gradient Bis-Tris gels and transferred onto PVDF membranes using an Invitrogen iblot two transfer device. After blocking of non-specific interactions using Li-Cor blocking reagent, membranes were incubated in PBS containing antibodies at the following concentrations: NKX2-1: (ab76013), 1:1000; PGC1α: (ab3242), 1:500. Protocol was repeated twice for a total of n = 3 with equivalent results.

## Acknowledgements

The authors would like to thank: Tim Mosbruger in the Bioinformatics Core at the Huntsman Cancer Institute (HCI); The Embryonic Stem Cell Core at the University of California, San Francisco (UCSF); The Gladstone Transgenic Gene-Targeting Core at UCSF; The Pathology cores at both UCSF and HCI; Brian Dalley and the the High Throughput Genomics Core at the Huntsman Cancer Center; and The FACS Facility at the Helen Diller Comprehensive Cancer Center at UCSF. We also thank Joseph Juan for assistance in tumor induction, Alex Jones and Eric Snyder for generously sharing the PCDH-FOXA1 construct pre-publication, Eric Snyder for helpful discussions and critical manuscript editing, Stephanie Correa for helpful discussions and equipment usage, and Holly Ingraham for generously sharing the pCDNA3-mLRH1 construct pre-publication. Dr. McMahon and colleagues gratefully acknowledge philanthropic support for this research from Five for the Fight.

## Additional information

### Competing interests

Martin McMahon: Reviewing editor, *eLife*. The other authors declare that no competing interests exist.

### Funding

| Funder | Grant reference number | Author |
| --- | --- | --- |
| National Cancer Institute | NCI R01_CA131261 | Martin McMahon |
| SASS | Postdoctoral Fellowship | J Edward van Veen Martin McMahon |
| University of California, San Francisco | Program for Breakthrough Biomedical Research | J Edward van Veen Martin McMahon |
| National Cancer Institute | 5T32CA108462-1 | J Edward van Veen Martin McMahon |

| National Cancer Institute | NCI P30_CA042014 | Martin McMahon |

The funders had no role in study design, data collection and interpretation, or the decision to submit the work for publication.

## Author contributions

J Edward van Veen, Conceptualization, Data curation, Software, Formal analysis, Supervision, Funding acquisition, Validation, Investigation, Visualization, Methodology, Writing—original draft, Project administration, Writing—review and editing; Michael Scherzer, Conceptualization, Data curation, Supervision, Investigation, Methodology, Writing—review and editing; Julia Boshuizen, Mollee Chu, Allison Landman, Shon Green, Conceptualization, Investigation, Writing—review and editing; Annie Liu, Investigation, Writing—review and editing; Christy Trejo, Martin McMahon, Conceptualization, Data curation, Funding acquisition, Investigation, Methodology, Project administration, Writing—review and editing

## Author ORCIDs

J Edward van Veen (iD) https://orcid.org/0000-0003-1798-3210
Martin McMahon (iD) https://orcid.org/0000-0003-2812-1042

## Ethics

Animal experimentation: All mouse work was done with the approval of the University of California IACUC under approval #AN089594. Mice were housed in microisolator cages on ventilated racks in an AAALAC accredited facility.

## Decision letter and Author response

Decision letter https://doi.org/10.7554/eLife.43668.049
Author response https://doi.org/10.7554/eLife.43668.050

# Additional files

## Supplementary files

• Transparent reporting form
DOI: https://doi.org/10.7554/eLife.43668.043

## Data availability

Sequencing data have been deposited in GEO under accession code GSE123126. All R scripts written for this study are available at GitHub (https://github.com/jevanveen/vanveen-elife; copy archived at https://github.com/elifesciences-publications/vanveen-elife).

The following dataset was generated:

| Author(s) | Year | Dataset title | Dataset URL | Database and Identifier |
|---|---|---|---|---|
| van Veen JE, Scherzer M, Boshuizen J, Chu M, Liu A, Landman A, Green S, McMahon M | 2018 | Mutationally-activated PI3'-kinase-α promotes de-differentiation of lung tumors initiated by the BRAFV600E oncoprotein kinase | https://www.ncbi.nlm.nih.gov/geo/query/acc.cgi?acc=GSE123126 | NCBI Gene Expression Omnibus, GSE123126 |

The following previously published dataset was used:

| Author(s) | Year | Dataset title | Dataset URL | Database and Identifier |
|---|---|---|---|---|
| Joshua D Campbell, Anton Alexandrov, Jaegil Kim, Jeremiah Wala, Alice H Berger, Chandra | 2016 | Distinct patterns of somatic genome alterations in lung adenocarcinomas and squamous cell carcinomas. | http://www.cbioportal.org/study?id=nsclc_tcga_broad_2016#summary | cBioPortal, nsclc_tcga_broad_2016 |

Sekhar Pedamallu,
Sachet A Shukla,
Guangwu Guo, An-
gela N Brooks,
Bradley A Murray,
Marcin Imielinski,
Xin Hu, Shiyun Ling,
Rehan Akbani,
Mara Rosenberg,
Carrie Cibulskis,
Aruna Ramachan-
dran, Eric A Collis-
son, David J
Kwiatkowski, Mi-
chael S Lawrence,
John N Weinstein,
Roel G W Verhaak,
Catherine J Wu,
Peter S Hammer-
man, Andrew D
Cherniack, Gad
Getz, Cancer Gen-
ome Atlas Research
Network, Maxim N
Artyomov, Robert
Schreiber, Ramas-
wamy Govindan,
Matthew Meyerson

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
