## [Decision Letter]

Thank you for submitting your article "Mutationally-activated PI3Kα promotes de-differentiation of lung tumors initiated by the BRAF^V600E^ oncoprotein kinase" for consideration by *eLife*. Your article has been reviewed by three peer reviewers, including Maureen Murphy as the Reviewing Editor and Reviewer #1, and the evaluation has been overseen by Jonathan Cooper as the Senior Editor. The following individual involved in review of your submission has agreed to reveal their identity: David Feldser (Reviewer #3).

The reviewers have discussed the reviews with one another and the Reviewing Editor has drafted this decision to help you prepare a revised submission.

Summary:

In this manuscript McMahon and colleagues have created an innovative new lung cancer model to lineage trace BRAF^V600E^-expressing cells by incorporating a tdTomato cDNA downstream of the endogenous Braf transcript. The tdTomato is separated from Braf during translation due to a P2A self-releasing peptide sequence. This new allele will be a valuable resource for the community. The new allele is used to determine the functional impact on cell differentiation state in BRAF^V600E^-driven lung adenoma/adenocarcinoma after expression of an oncogenic form of PIK3CA. The focus is born out of an unbiased observation from RNA-Seq analysis on sorted tdTomato positive tumor cells where BRAF/PIK3CA tumors have a strong suppression of alveolar type II (ATII) pneumocyte gene expression compared to BRAF only tumor cells.

Key insights:

1) Oncogenic PIK3CA expression suppresses ATII marker expression in a subset of tumor cells; surprisingly, this occurs without changes in the ATII lineage specifying transcription factors NKX2-1, FOXA1, or FOXA2.

2) Informatics approaches identify PGC1α as a key transcriptional regulator that enforces ATII cell identity. PGC1α expression is suppressed in BRAF/PIK3CA cells that lose ATII lineage marks and BRAF^V600E^/+; PGC1α-/- tumors globally lose ATII marks. This widespread effect contrasts with the heterogeneous effect that is observed in the BRAF/PIK3CA model.

3) PGC1α and a nuclear receptor (NR5A2) can drive ATII gene expression in heterologous reporter assays.

4) PGC1α and NKX2-1 are found in a complex in immortalized mouse epithelial cells suggesting that they operate together naturally to enforce ATII cell fate.

5) The authors demonstrate that the PIK3CA-induced effects on lost lineage marker expression is conserved in KRAS-driven disease as well, suggesting that the effects of oncogenic PIK3CA might apply more broadly in lung adenocarcinoma patients.

The insights presented here are novel. They help pinpoint mechanisms of action for a highly important oncoprotein within a disease setting of high clinical need. The mechanisms highlighted are significant because despite an enormous amount of effort to study PIK3CA cancer driving mutations over multiple decades, the authors identify a novel mechanism of action for oncogenic PIK3CA signaling to suppress PGC1α-controlled ATII cell fate enforcement.

Essential revisions:

The following additional experiments, when feasible, would strengthen the main conclusions of the manuscript.

1) The authors primarily define the requirement for PGC1α in vivo, which is important. However, this finding would be strengthened if it could be supported by acute, in vitro studies if possible. Based on the model described, if there are KP NSCLC cells that exist that still retain NKX2.1/ATII marker expression it would be helpful if the authors could genetically suppress PGC1α and show a decrease in ATII markers. Conversely, if these genes are not expressed (in available cell lines) perhaps the 4 transcription factor cocktail would increase ATII marker expression in these cancer cell lines. A positive result would be more compelling than the reporter construct experiments.

2) Figure 2C shows that an ATII signature is significantly depleted in BRAF/PIK3CA tumor cells, and the authors focus on this result. However, this figure also shows a profound enrichment for ATI genes that were identified in the same study highlighted by the authors for the ATII gene set. Additionally, both club cell and ciliated cell signatures appear to be depleted upon activation of PI3K signaling. Accordingly, the data suggest that ATII lineage is not specifically depleted upon PI3K activation, but rather that a widespread cell altering phenotype may be taking place. Further exploration of this possibility would yield a better understanding of the extent of transdifferentiation and cell of origin in the authors findings. Can the authors perform immunofluorescence staining for ATI, Club, and ATII markers at several time points after tumor initiation (including very early times, ~1-2 weeks). Though it is stated in Figure 2A that Ad5-SPC-CRE is used which should restrict CRE expression to ATII cells, it would be a nice observation to identify how ATI and Club cell transcripts are changing over time since obviously they are detected in the RNA-Seq analysis. Some explanation for these changing gene sets is required. Can the authors comment as to how much overlap there is between these signatures?

3) In Figure 6D the authors use luciferase assays to address whether PGC1α, NR5A2, or PGC1α/NR5A2 combo is critical for mediating FOXA2- and NKX2-1-dependent transcription of surfactant proteins A, B and C. However, based on these assays the NR5A2/PGC1α combo is sufficient to drive high expression of SFTPC and this is independent of NKX2-1 and FOXA2. In fact, NKX2-1 and FOXA2 seems entirely dispensable for expression of SFTPC reporter. In contrast, the NR5A2/PGC1α combination synergistically enhances FOXA2- and NKX2-1-dependent SFTPA and SFTPB expression. Reconciliation of these findings might be aided by an in vivo experiment, such as a ChIP-qPCR experiment looking at PGC1α, NR5A2, NKX2-1 and FOXA2 binding to Sftpa, Sftpb, Sftpc promoters in lung tumors or tumor-derived cell lines with or without mutant PI3K; this would be more convincing than the transient reporter assays.

---

## [Author Response]

Essential revisions:The following additional experiments, when feasible, would strengthen the main conclusions of the manuscript.1) The authors primarily define the requirement for PGC1α in vivo, which is important. However, this finding would be strengthened if it could be supported by acute, in vitro studies if possible. Based on the model described, if there are KP NSCLC cells that exist that still retain NKX2.1/ATII marker expression it would be helpful if the authors could genetically suppress PGC1α and show a decrease in ATII markers. Conversely, if these genes are not expressed (in available cell lines) perhaps the 4 transcription factor cocktail would increase ATII marker expression in these cancer cell lines. A positive result would be more compelling than the reporter construct experiments.

We agree that recapitulating the effects that we have observed in vivo in an appropriate lung cancer cell line would strengthen our conclusions. Unfortunately, we can find no lung cancer cell line that maintains the expression of markers of normal AT2 pneumocyte differentiation. First, it is reported that isolation and culture of mouse AT2 cells leads to a rapid decrease in AT2 marker expression, and transdifferentiation into AT1 pneumocytes (Demaio L, Tseng W, Balverde Z, et al. Characterization of mouse alveolar epithelial cell monolayers. Am J Physiol Lung Cell Mol Physiol. 2009;296(6):L1051–L1058. doi:10.1152/ajplung.00021.2009). Moreover, we (and others) have tried isolating and culturing AT2 pneumocytes from the “Immortomouse” (*H-2K^b^::LargeT^tsA58^*) but with no success. We have made substantial efforts to employ the reviewers’ suggestion regarding transfecting the four transcription factor cocktail into LUAD cell lines, but we were unable to observe any increase in AT2 marker expression. As an alternative approach to address this we also attempted to pharmacologically inhibit PI3K signaling in three human lung cancer cell lines with predicted activation of PI3’-lipid signaling (NCI-H358, NCI-1299, NCI-H1975), and then assessed expression of AT2 markers. While we were able to inhibit phosphorylation of AKT in these experiments, we did not detect any increase in the expression of SFTPA, ETV5, or LYZ, perhaps due to the complicated mutational landscapes of these cell lines. Consequently, while we agree that these experiments would indeed strengthen this study, they have not been readily tractable even within the context of the extended time-frame of the revisions of this manuscript.

2) Figure 2C shows that an ATII signature is significantly depleted in BRAF/PIK3CA tumor cells, and the authors focus on this result. However, this figure also shows a profound enrichment for ATI genes that were identified in the same study highlighted by the authors for the ATII gene set. Additionally, both club cell and ciliated cell signatures appear to be depleted upon activation of PI3K signaling. Accordingly, the data suggest that ATII lineage is not specifically depleted upon PI3K activation, but rather that a widespread cell altering phenotype may be taking place. Further exploration of this possibility would yield a better understanding of the extent of transdifferentiation and cell of origin in the authors findings. Can the authors perform immunofluorescence staining for ATI, Club, and ATII markers at several time points after tumor initiation (including very early times, ~1-2 weeks). Though it is stated in Figure 2A that Ad5-SPC-CRE is used which should restrict CRE expression to ATII cells, it would be a nice observation to identify how ATI and Club cell transcripts are changing over time since obviously they are detected in the RNA-Seq analysis. Some explanation for these changing gene sets is required. Can the authors comment as to how much overlap there is between these signatures?

Based on the reviewers’ suggestion we have added a new figure (Figure 5) exploring the expression of AT1 pneumocyte markers in BRAF^V600E^/PI3Kα^H1047R^- versus BRAF^V600E^-driven lung tumors. We now report on finding unanticipated patterns of AT1 marker expression, which is explored in some depth in the revised manuscript. We extend this finding with analysis of transmission electron micrographs from BRAF^V600E^-driven lung tumors. Together these analyses confirm and expand upon our previously made argument that MAPK and PI3K signaling cooperate to promote de-differentiation of AT2 cells in the process of lung tumorigenesis.

3) In Figure 6D the authors use luciferase assays to address whether PGC1α, NR5A2, or PGC1α/NR5A2 combo is critical for mediating FOXA2- and NKX2-1-dependent transcription of surfactant proteins A, B and C. However, based on these assays the NR5A2/PGC1α combo is sufficient to drive high expression of SFTPC and this is independent of NKX2-1 and FOXA2. In fact, NKX2-1 and FOXA2 seems entirely dispensable for expression of SFTPC reporter. In contrast, the NR5A2/PGC1α combination synergistically enhances FOXA2- and NKX2-1-dependent SFTPA and SFTPB expression. Reconciliation of these findings might be aided by an in vivo experiment, such as a ChIP-qPCR experiment looking at PGC1α, NR5A2, NKX2-1 and FOXA2 binding to Sftpa, Sftpb, Sftpc promoters in lung tumors or tumor-derived cell lines with or without mutant PI3K; this would be more convincing than the transient reporter assays.

After considering this critique we agree that it is odd that in our system NKX2-1 appeared dispensable for transactivation of the SFTPC promoter and so we more carefully examined our luciferase assay data. Indeed, although not visible in the original Figure 6D referenced, we did see a modest but significant (Wilcoxon rank sum test, p<.05) induction of the Sftpc promoter by NKX2-1 expression – see Author response image 1. We have searched for a way to perform the experiments suggested regarding ChIP of NR5A2 at the Sftpc promoter, but have encountered trouble harvesting enough material from tumor-bearing lungs by FACS for anything other than RNA sequencing. We are keen to perform the suggested experiments as a continuation of these studies more completely exploring the role of PGC1α in lung tumor progression. However, we would respectfully suggest that these experiments might be considered outside of the scope of the current manuscript.
